# FUSECHAT: KNOWLEDGE FUSION OF CHAT MODELS

## ABSTRACT

While training large language models (LLMs) from scratch can indeed lead to models with distinct capabilities and strengths, it incurs substantial costs and may lead to redundancy in competencies. Knowledge fusion aims to integrate existing LLMs of diverse architectures and capabilities into a more potent LLM through lightweight continual training, thereby reducing the need for costly LLM development. In this work, we propose a new framework for the knowledge fusion of chat LLMs through two main stages, resulting in FUSECHAT. Firstly, we conduct pairwise knowledge fusion on source chat LLMs of varying structures and scales to create multiple target LLMs with identical structure and size via lightweight fine-tuning. During this process, a statistics-based token alignment approach is introduced as the cornerstone for fusing LLMs with different structures. Secondly, we merge these target LLMs within the parameter space, where we propose a novel method for determining the merging coefficients based on the magnitude of parameter updates before and after fine-tuning. We implement and validate FUSECHAT using six prominent chat LLMs with diverse architectures and scales, including OpenChat-3.5-7B, Starling-LM-7B-alpha, NH2-SOLAR-10.7B, InternLM2-Chat-20B, Mixtral-8x7B-Instruct, and Qwen-1.5-Chat-72B. Experimental results on two instruction-following benchmarks, AlpacaEval 2.0 and MT-Bench, demonstrate the superiority of FUSECHAT-7B over baselines of various sizes. Our model is even comparable to the larger Mixtral-8x7B-Instruct and approaches GPT-3.5-Turbo-1106 on MT-Bench as Figure 1(b).

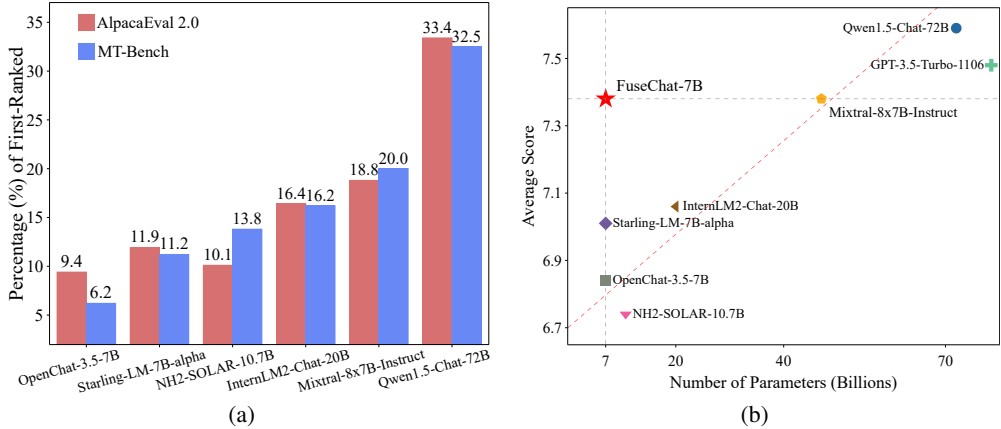

(a)                                     (b)

Figure 1: Demonstration (left) of distinct strengths of existing chat LLMs and comparison (right) between FUSECHAT-7B and baseline LLMs. While the left figure plots the percentage of first-ranked responses of each LLM as measured by PairRM (Jiang et al., 2023) on AlpacaEval 2.0 and MT-Bench, the right shows that FUSECHAT-7B achieves comparable performance to Mixtral-8x7B and approaches GPT-3.5 on MT-Bench. The red dashed line is linearly fitted from data points of all chat LLMs except FUSECHAT-7B.

# 1 INTRODUCTION

Large language models (LLMs) such as GPT (Brown et al., 2020) and LLaMA (Touvron et al., 2023) series have demonstrated remarkable success across a wide range of natural language processing (NLP) tasks. Currently, it has become prevalent and imperative for individuals and corporations to build their own LLMs. However, the computational resources and time costs associated with LLM development remain prohibitively high. Furthermore, despite the structural and functional differences among LLMs, they often exhibit similar capabilities across various tasks. Therefore, besides training an LLM from scratch, another option is to combine the distinct advantages of existing LLMs into a more potent LLM, which is termed *knowledge fusion of LLMs* (Wan et al., 2024). Figure 1(a) illustrates the results of our preliminary experiment conducted on AlpacaEval 2.0 and MT-Bench, where we plot the percentage of questions each LLM answers best (measured by PairRM (Jiang et al., 2023)) among six prominent chat LLMs. These established LLMs, regardless of their potency, exhibit distinct strengths. Therefore, knowledge fusion not only reduces the developmental costs of creating a new LLM but also has the potential to integrate the diverse strengths of existing models.

The endeavor to integrate the capabilities of multiple models has been a long-standing pursuit. For example, ensemble methods (Littlestone & Warmuth, 1994; Jiang et al., 2023) directly aggregate the outputs of multiple models to enhance prediction performance and robustness. However, this approach requires maintaining multiple trained models during inference, which is inefficient for LLMs due to their substantial memory and inference time requirements. Another approach is to directly merge several neural networks into a single network through arithmetic operations in the parameter space (Gupta et al., 2020), whereas this approach typically assumes uniform network architectures and requires manually-tuned (Wortsman et al., 2022; Yadav et al., 2024) or automatically-learned (Matena & Raffel, 2022; Jin et al., 2023) coefficients to merge the parameters of different neural networks. In contrast, knowledge fusion (Wan et al., 2024) seeks to integrate the capabilities of multiple LLMs, irrespective of their architectures, into a single LLM through lightweight continual training. This process essentially embodies a traditional multi-teacher knowledge distillation procedure (You et al., 2017), but faces new challenges such as token alignment and fusion strategies across different LLMs.

In this study, we introduce a fuse-and-merge framework to extend the fusion of LLMs to chat-based LLMs[1] with diverse architectures and scales through two stages, resulting in FUSECHAT. Firstly, we conduct pairwise knowledge fusion for source chat LLMs to generate multiple target LLMs of identical structure and size. To achieve this, we first select a pivot LLM and perform token alignment, followed by knowledge fusion between the pivot and each of the remaining LLMs. These target LLMs are expected to inherit the strengths of source chat LLMs through knowledge transfer during lightweight fine-tuning. Secondly, these target LLMs are merged within the parameter space, where we introduce a novel method called SCE to determine the merging coefficients based on the magnitude of parameter updates before and after fine-tuning. Moreover, SCE allocates parameter matrix-level coefficients that enable the merging at a fine-grained granularity without additional training efforts.

FUSECHAT offers superior potential compared to FUSELLM (Wan et al., 2024). Firstly, while FUSELLM limits its exploration to source LLMs of the same size as the target LLM, FUSECHAT broadens the scope by incorporating six source LLMs with varying scales. This allows for greater adaptability to the fusion of diverse chat LLMs. Secondly, the framework of FUSELLM does not seamlessly support the inclusion of new source LLMs as it requires the combination of distribution matrices from all source LLMs during continual training. In contrast, integrating a new source LLM in FUSECHAT is plug-and-play, requiring only obtaining a target LLM from the new source LLM and merging it with the existing FUSECHAT. Thirdly, compared to many-to-one knowledge fusion, pairwise fusion empirically mitigates the challenges of knowledge distillation from source LLMs.

To verify the effectiveness of FUSECHAT, we implemented FUSECHAT-7B using six prominent open-source chat LLMs: OpenChat-3.5-7B (Wang et al., 2024a), Starling-LM-7B-alpha (Zhu et al., 2023), NH2-SOLAR-10.7B (Kim et al., 2023), InternLM2-Chat-20B (Cai et al., 2024), Mixtral-8x7B-Instruct (Jiang et al., 2024), and Qwen-1.5-Chat-72B (Bai et al., 2023). Experimental results on two representative instruction-following benchmarks, AlpacaEval 2.0 (Dubois et al., 2024b) and MT-Bench (Zheng et al., 2024), demonstrate the superiority of FUSECHAT-7B across a broad spectrum of chat LLMs at 7B, 10B, and 20B scales. Moreover, we validated the proposed token alignment method and the SCE merging method through a series of analytical experiments.

---

[1]We will refer to "chat-based LLMs" simply as "chat LLMs" for brevity.

## 2 RELATED WORK

**Model Fusion**   Combining the capabilities of diverse models has been a long-standing objective. Existing approaches to model fusion mainly fall into three categories. Firstly, traditional *model ensemble* techniques combine the outputs of multiple models by weighted averaging (Littlestone & Warmuth, 1994) or majority voting (Monteith et al., 2011) to enhance overall system performance. Recently, Jiang et al. (2023) introduced a sequence-level ensemble framework for LLMs, which first conducts pairwise comparisons to rank the outputs of LLMs and then employs another LLM to consolidate the top-ranked candidates into an improved output. In addition to the sequence-level ensemble, Ding et al. (2024) blended multiple LLMs using a token-level gating mechanism on the output logits. To avoid additional training during ensemble, Mavromatis et al. (2024) leveraged the perplexity of different LLMs over input prompts to determine the importance of each model.

Secondly, *model merging* facilitates the fusion of models of identical structure and scale within the parameter space. Wortsman et al. (2022) combined multiple models, obtained by fine-tuning a model on the same dataset but with distinct strategies, through linear averaging. Matena & Raffel (2022) enhanced simple weighted average by incorporating Fisher Information Matrix (Fisher, 1922) to determine the significance of individual model parameter. Jin et al. (2023) performed merging by addressing an optimization problem that minimizes the L2 distance between merged and individual models, and conducting a closed-form solution. Although these methods can automatically compute merging coefficients, they necessitate either forward or backward propagation using additional data, making model merging compute-inefficient and memory-intensive. Ilharco et al. (2023) and Zhang et al. (2023) conducted simple arithmetic operations on the task vectors or LoRA (Hu et al., 2022) modules of different models, thereby enhancing multi-task ability and domain generalization. To mitigate parameter interference, Yu et al. (2023) and Yadav et al. (2024) introduced sparsification techniques that trim redundant values from task vectors prior to model merging. Moreover, Kim et al. (2023) and Akiba et al. (2024) advanced the field by merging multiple LLMs across both parameter and data flow spaces, yielding a merged LLM with up-scaled depth and superior performance.

Thirdly, *mixture of experts* (MoEs) combines specialized expert modules with a sparsely activated mechanism (Fedus et al., 2022), presenting another venue for model fusion. Komatsuzaki et al. (2023) first proposed initializing a sparse MoEs module using multiple copies from a dense checkpoint. To integrate multiple domain experts, Sukhbaatar et al. (2024) trained multiple domain-specific LLMs from a seed LLM separately and then used feed-forward networks on top of these dense experts to instantiate a sparse MoEs module, followed by further fine-tuning to learn token-level routing. Similarly, Feng et al. (2024) trained multiple domain-specific LoRA (Hu et al., 2022) modules as experts and combined these domain experts using an explicit sequence-level routing strategy.

Lastly, FUSELLM (Wan et al., 2024) introduces another paradigm for the fusion of LLMs with structural differences. This approach builds upon knowledge distillation and leverages the probabilistic distribution matrices generated by source LLMs to transfer collective knowledge into a target LLM. Unlike model ensembles and MoEs, knowledge fusion does not require the parallel deployment of multiple models (experts). Furthermore, compared to model merging, which only applies to models with identical architectures, FUSELLM allows for the fusion of LLMs with different architectures.

**Knowledge Distillation**   Knowledge fusion essentially performs knowledge distillation to transfer knowledge from source LLMs to a target LLM. Knowledge distillation (Hinton et al., 2015) aims to train a small student model guided by one or more larger teacher models. Previous studies primarily focus on training a student model to mimic the teacher's behavior in text classification tasks, by replicating the teacher's output logits (Sanh et al., 2019; Turc et al., 2019), as well as hidden states (Sun et al., 2019; Jiao et al., 2020) and relations (Wang et al., 2020). In the realm of generative models, prevailing approaches maximize the log-likelihood of the student on the distributions (Khanuja et al., 2021; Gu et al., 2024; Agarwal et al., 2024) or sequences (Kim & Rush, 2016; Peng et al., 2023) generated by the teacher model. This paradigm can be extended to accommodate multiple teachers by either averaging the distributions (You et al., 2017) or blending the sequences (Wang et al., 2024a).

Compared to vanilla knowledge distillation, knowledge fusion of LLMs faces new challenges. Firstly, due to the differences in tokenization among various LLMs, token alignment is essential for transferring knowledge from source to target LLMs. Secondly, when dealing with distributions generated from multiple source LLMs, the fusion function becomes crucial for optimally integrating their distributions. Thirdly, to leverage the unique advantages of different LLMs, it is necessary and challenging to create a compact knowledge fusion dataset that is diverse in capabilities and domains.

Figure 2: Overview of FUSECHAT in comparison with FUSELLM (Wan et al., 2024). Distinct animal icons symbolize different LLMs, where each species and size indicate a unique architecture and scale, respectively.

## 3    FUSECHAT

### 3.1    OVERVIEW

Figure 2 presents an overview of our FUSECHAT in comparison with FUSELLM (Wan et al., 2024). The FUSECHAT framework consists of two main stages: fuse and merge. In the *fuse* stage, pairwise knowledge fusion is conducted on source chat LLMs[2] to derive multiple target LLMs with identical structure and size. This process begins by selecting a pivot LLM, followed by performing knowledge fusion between the pivot and each remaining LLM. In the *merge* stage, these target LLMs are combined within the parameter space, where we introduce a novel method to determine the merging coefficients based on the magnitude of parameter updates before and after fine-tuning.

Specifically, considering $K$ source LLMs $\{\mathcal{M}_i^s\}_{i=1}^K$ with varying architectures and scales, FUSECHAT first specifies one of the source LLMs, $\mathcal{M}_v^s$, as the pivot and then applies pairwise knowledge fusion to obtain $(K-1)$ target LLMs, $\{\mathcal{M}_j^t\}_{j=1}^{K-1}$, which share the same architecture and initialized weights as the pivot LLM. The selection of the pivot depends on the desired structure and scale for the target LLMs, while also considering the capabilities and performance of a candidate LLM.

To perform pairwise knowledge fusion, FUSECHAT prompts these source LLMs using a supervised fine-tuning dataset $\mathcal{D} = \{I_i, R_i\}_{i=1}^M$ to showcase their inherent knowledge by responding to each instruction in $\mathcal{D}$. Token alignment (Fu et al., 2023; Wan et al., 2024) between the source LLMs and the pivot is then conducted to properly map the resulting probabilistic distribution matrices. These distribution matrices are subsequently used for pairwise knowledge fusion (Wan et al., 2024) through lightweight fine-tuning to obtain $(K-1)$ target LLMs. Following this, the target LLMs are merged in the parameter space to yield the final fused LLM $\mathcal{M}^f$. To incorporate fine-grained advantages of target LLMs, we introduce a new merging method named SCE to obtain the merging coefficients based on *selection*, *calculation*, and *erasure* on the task vectors (Ilharco et al., 2023) which represent variation of model weights before and after fine-tuning. SCE enables the automatic allocation of parameter matrix-level merging coefficients, facilitating the merging of LLMs at a finer granularity.

### 3.2    PRELIMINARIES

Given an instruction $I_i$ and the corresponding response $R_i$ of length $N$ from the fine-tuning dataset $\mathcal{D}$, we use $R_{i,<t} = (r_{i,1}, r_{i,2}, \ldots, r_{i,t-1})$ to represent the sequence preceding the $t$th token in the response. The supervised fine-tuning (SFT) objective for a pre-trained language model parameterized by $\theta$ is defined as minimizing the following negative log-likelihood:

$$\mathcal{L}_{\text{SFT}} = -\mathbb{E}_{(I_i, R_i)\sim\mathcal{D}} \left[ \sum_{t \leq N} \log p_\theta(r_{i,t}|R_{i,<t}, I_i) \right], \tag{1}$$

where $p_\theta(r_{i,t}|R_{i,<t}, I_i)$ is the model's predicted probability for the $t$th token $r_{i,t}$ in $R_i$ given the instruction and preceding tokens in the response.

---

[2]We will use "source chat LLMs" and "source LLMs" interchangeably when there is no ambiguity.

## 3.3 PAIRWISE KNOWLEDGE FUSION

To facilitate the description of pairwise knowledge fusion, we reframe the above token-level view into a matrix format. Specifically, for each instruction $I_i$, we transform the token-level predictions into a probabilistic distribution matrix, $\mathbf{P}_i^\theta \in \mathbb{R}^{N \times V}$, where $V$ denotes the vocabulary size. The distribution matrix is assumed to reflect certain inherent knowledge of the language model in responding to the instruction (Wan et al., 2024). Consequently, different probabilistic distribution matrices obtained from different chat LLMs can be used to represent the diverse knowledge embedded within these models. Based on this assumption, FUSECHAT performs pairwise knowledge fusion by fine-tuning the target LLMs, initialized from the pivot, using the probabilistic distribution matrices.

**Model Fusion** For each instruction $I_i$ in $\mathcal{D}$, we first feed it into the $K$ source chat LLMs to obtain a set of probabilistic distribution matrices, denoted as $\{\mathbf{P}_i^{\theta_j}\}_{j=1}^K$, where $\theta_j$ represents the parameters of the $j$th chat LLM. Since these LLMs may employ different tokenizers, token alignment is necessary to properly map their probabilistic distribution matrices (Fu et al., 2023; Wan et al., 2024). Then, pairwise knowledge fusion is conducted between the pivot LLM and each of the remaining source LLMs. To achieve this, we denote the probabilistic distribution matrix generated by the pivot LLM as $\mathbf{P}_i^{\theta_v}$ and merge it with each $\mathbf{P}_i^{\theta_j}|_{j \neq v}$ to obtain a set $\{\mathbf{P}_i^j\}_{j=1}^{K-1}$ of fused matrices as follows:

$$\mathbf{P}_i^j = \mathbb{F}\text{usion}(\mathbf{P}_i^{\theta_v}, \mathbf{P}_i^{\theta_j})|_{j \neq v}, \tag{2}$$

where $\mathbb{F}\text{usion}(\cdot)$ represents the fusion function that merges two matrices. The resulting matrix $\mathbf{P}_i^j$ is seen as a representation of the collective knowledge and distinctive strengths of the two source LLMs. Among various fusion strategies, this work employs minimum cross-entropy (MinCE) (Wan et al., 2024) as the fusion function, which empirically performs the best.

After that, we enforce alignment between the prediction of each target LLM $\mathcal{M}_j^t$ and the corresponding fused representation matrices $\mathbf{P}_i^j$. We use $\mathbf{Q}_i^{\phi_j}$ to represent the output distribution matrix of target LLM $\mathcal{M}_j^t$ for instruction $I_i$ and define the fusion objective for training each target LLM as follows:

$$\mathcal{L}_{\text{Fusion}} = -\mathbb{E}_{(I_i, R_i) \sim \mathcal{D}} \left[ \mathbb{H}(\mathbf{P}_i^j || \mathbf{Q}_i^{\phi_j}) \right], \tag{3}$$

where $\mathbb{H}(\cdot || \cdot)$ represents the cross entropy between two probabilistic distribution matrices.

The overall training objective for each pairwise knowledge fusion consists of a weighted combination of the supervised fine-tuning objective $\mathcal{L}_{\text{SFT}}$ and the fusion objective $\mathcal{L}_{\text{Fusion}}$ as follows:

$$\mathcal{L} = \lambda \mathcal{L}_{\text{SFT}} + (1 - \lambda) \mathcal{L}_{\text{Fusion}}. \tag{4}$$

**Token Alignment** Token alignment aims to address the mappings of probabilistic distribution matrices $\{\mathbf{P}_i^{\theta_j} \in \mathbb{R}^{N \times V}\}_{j=1}^K$ generated by different source LLMs for a given instruction $I_i$. Therefore, the alignment involves two dimensions of the matrices: the sequence dimension for the tokenized response and the distribution dimension for the probabilistic distributions. In the sequence dimension, we follow previous works (Fu et al., 2023; Wan et al., 2024) and adopt dynamic programming to recursively minimize the total cost of editing the tokens from a source LLM to align them with the pivot LLM. This process may result in 1-1, 1-n, and n-1 mappings, as shown in Figure 7. In the distribution dimension, Fu et al. (2023) focused on aligning distributions based on the exact match (EM) between tokens in source and target distributions, which restricts the alignment to only 1-1 mappings and may result in too many unmatched tokens. Wan et al. (2024) relaxed the EM constraint by aligning the distributions based on the minimum edit distance (MinED) between tokens in the vocabularies of source and target LLMs. While this approach improves the mapping success rate and expands to 1-n mappings, it ignores n-1 mappings and may introduce many misalignments.

In this work, we propose an enhanced token alignment strategy that utilizes mapping statistics (MS) from the sequence dimension as the criteria for alignment in the distribution dimension. We construct a global statistical matrix, where each column represents the frequency of mappings from a pivot token to all potential source tokens, derived from sequence-dimensional token alignments. In the case of 1-1 and 1-n mappings, we align the distributions based on the maximum mapping frequency in the respective columns of the statistical matrix for each pivot token in the distribution. For n-1 mappings,

we first calculate a weighted average of the source tokens' distributions according to their mapping frequencies in the statistical matrix to obtain a merged distribution. This merged distribution is then aligned to the pivot distribution similar to the procedure employed for 1-1 mappings. As illustrated in Figure 7, this approach better reflects the token mapping statistics in the dataset, thereby preserving significant information in the aligned distribution matrices while minimizing alignment errors.

## 3.4 MODEL MERGING

Since the target LLMs $\{\mathcal{M}_j^t\}_{j=1}^{K-1}$ resulting from pairwise knowledge fusion share identical architecture and scale while possessing diverse advantages and capabilities learned from the source LLMs, we further merge them in the parameter space to obtain the final fused LLM $\mathcal{M}^f$. To ensure the adaptability and scalability of FUSECHAT, it is essential to maintain the simplicity of the merging strategy. Primarily, the calculation of merging coefficients should be automated, obviating the complex hyperparameter tuning. Secondly, the merging procedure should not demand forward or backward propagation over additional data, which is computationally inefficient and memory-intensive.

As described in Algorithm 1, we propose a novel merging method named SCE (*select*, *calculate*, and *erase*) for parameter matrix-level merging. Analogous to task vectors (Ilharco et al., 2023), we first define fusion vectors $\{\delta_j\}_{j=1}^{K-1}$ (Eq. 5) as the direction and magnitude of weight updates from pivot LLM $\mathcal{M}_v^s$ to target LLMs $\{\mathcal{M}_j^t\}_{j=1}^{K-1}$ during model fusion. For each parameter matrix unit in target LLMs, we derive the merged weights using fusion vectors through a three-step process.

(1) **Select**: During the pairwise knowledge fusion, target LLMs dynamically evolve their parameters to incorporate the advantages of their corresponding source LLMs. Fusion vectors for each parameter matrix unit with substantial variations across different target LLMs are supposed to signify distinctive and significant strengths. Therefore, we first select the top $\tau\%$ elements from each parameter matrix-level fusion vector $\{\delta_{j,m}\}_{j=1}^{K-1}$ with high variance across multiple target LLMs, resulting in $\{\hat{\delta}_{j,m}\}_{j=1}^{K-1}$ (Eq. 6). (2) **Calculate**: We then calculate the sum of squares of elements in $\hat{\delta}_{j,m}$ and obtain a matrix-level merging coefficient for each target LLM as $\eta_{j,m} = \frac{\sum \hat{\delta}_{j,m}^2}{\sum_j \sum \hat{\delta}_{j,m}^2}$. (3) **Erase**: Each parameter may exhibit conflicting signs across fusion vectors from different target LLMs, which could cause interference during model merging (Yadav et al., 2024). Thus, for each parameter we sum its values in $\{\hat{\delta}_{j,m}\}_{j=1}^{K-1}$ across target LLMs and erase elements with minority directions (Eq. 8). Finally, the filtered $\{\delta'_{j,m}\}_{j=1}^{K-1}$ are merged based on the calculated coefficients, and added to the pivot LLM's parameters (Eq. 9).

---

**Algorithm 1** SCE Procedure

**Input:** target LLMs parameters $\{\phi_j\}_{j=1}^{K-1}$, pivot LLM parameters $\theta_v$, threshold $\tau$.
**Output:** merged LLM parameters $\Phi$
▷ Create fusion vectors
$$\{\delta_j\}_{j=1}^{K-1} = \{\phi_j - \theta_v\}_{j=1}^{K-1} \qquad (5)$$
▷ Calculate parameter matrix-level merging coefficients
**for** $\{\delta_{j,m}\}_{j=1}^{K-1} \in \{\delta_j\}_{j=1}^{K-1}$ **do**
  ▷ Step 1: Select salient elements
$$\{\hat{\delta}_{j,m}\}_{j=1}^{K-1} = \text{Select}(\{\delta_{j,m}\}_{j=1}^{K-1}, \tau) \quad (6)$$
  ▷ Step 2: Calculate coefficients
$$\{\eta_{j,m}\}_{j=1}^{K-1} = \text{Calculate}(\{\hat{\delta}_{j,m}^2\}_{j=1}^{K-1}) \quad (7)$$
  ▷ Step 3: Erase minority elements
$$\{\delta'_{j,m}\}_{j=1}^{K-1} = \text{Erase}(\{\hat{\delta}_{j,m}\}_{j=1}^{K-1}) \qquad (8)$$
  ▷ Update merged LLM parameters
$$\Phi_m = \theta_{v,m} + \sum_{j=1}^{K-1} \eta_{j,m} \delta'_{j,m} \qquad (9)$$
**end**
**return** $\Phi$

---

## 3.5 DISCUSSIONS

The reasons why FUSECHAT adopts pairwise rather than many-to-one knowledge fusion as FUSELLM (Wan et al., 2024) are twofold. Firstly, directly fusing all the source LLMs proves to be difficult, as evidenced by the results of OpenChat-3.5-7B Multi in Table 1. Instead, FUSECHAT adopts a fuse-and-merge strategy, wherein the fusing stage employs pairwise knowledge fusion between the pivot LLM and other source LLMs, which reduces the difficulty of model fusion. Secondly, FUSECHAT offers superior scalability compared to FUSELLM. The framework of FUSELLM requires the combination of distribution matrices from all source LLMs during continual training, which does not easily support the inclusion of new LLMs. In contrast, FUSECHAT supports plug-and-play

integration of new source LLMs at any scale. This requires only obtaining a target LLM by fusing the new source LLM with the pivot, and then merging it with the existing version of FUSECHAT.

## 4 EXPERIMENTS

In our experiments, we explore the fusion of chat LLMs with diverse architectures and scales. Specifically, we conduct experiments using six representative chat LLMs as the source LLMs, including OpenChat-3.5-7B (Wang et al., 2024a), Starling-LM-7B-alpha (Zhu et al., 2023), NH2-SOLAR-10.7B (Kim et al., 2023), InternLM2-Chat-20B (Cai et al., 2024), Mixtral-8x7B-Instruct (Jiang et al., 2024), and Qwen-1.5-Chat-72B (Bai et al., 2023). As for the pivot LLM, which also serves as the starting point for the target LLMs, we opt for OpenChat-3.5-7B due to its balanced scale and performance. To begin, we first apply pairwise knowledge fusion (Section 3.3) to create five distinct target LLMs with the same structure. These target LLMs are then merged using the SCE method (Section 3.4), resulting in the final FUSECHAT-7B.

### 4.1 EXPERIMENTAL SETUP

**Training Dataset**  To leverage the strengths of source LLMs during knowledge fusion while alleviating catastrophic forgetting, we curate a high-quality dataset named FUSECHAT-MIXTURE from two different sources. First, 50% of the training instances are sampled from the dataset used by the pivot LLM, OpenChat-3.5-7B. Second, we gather the remaining training instances, which have not been encountered by the pivot LLM, from open-source communities. These two sources result in a corpus comprising approximately 95,000 dialogues across spanning various domains. For further details on FUSECHAT-MIXTURE, please refer to Appendix C.

**Training Details**  In all experiments, we train the target LLMs using a batch size of 128 and a maximum length of 2048 on a single node with 8x80GB NVIDIA A800 GPUs for three epochs, which takes approximately 9 hours. The models are optimized using the AdamW (Loshchilov & Hutter, 2019) optimizer with $\beta_1 = 0.9$ and $\beta_2 = 0.999$. We use a weight decay of 0.0 and gradient clipping of 1.0. A cosine learning rate schedule is employed, with a maximum learning rate of 5e-6 and a warmup ratio of 0.03. We empirically set the combination weight $\lambda$ in Eq. 4 to 0.9 and the rationale behind the value of $\lambda$ is detailed in Appendix H. Our training framework is implemented based on the HuggingFace Transformers (Wolf et al., 2020).

**Evaluation**  We assess the performance of FUSECHAT-7B on two representative benchmarks to evaluate its ability to follow instructions and engage in conversations effectively. The first benchmark, AlpacaEval 2.0 (Dubois et al., 2024b), comprises 805 instructions across five test subsets. It compares the Win Rate and Length-Controlled Win Rate (LC Win Rate) (Dubois et al., 2024a) of a model against GPT-4. We employ the default settings and utilize GPT-4 (GPT-4-1106-Preview) to evaluate the quality of generated responses. The second benchmark, MT-Bench (Zheng et al., 2024), consists of 80 multi-turn dialogues spanning various domains including writing, roleplay, reasoning, math, coding, extraction, STEM, and humanities. Originally, GPT-4 (GPT-4-0613) was used as the evaluator, providing a scalar score ranging from 1 (lowest) to 10 (highest) for each generated response. However, due to inaccuracies in the reference responses, we adopt an updated version, GPT-4-0125-Preview, as per the latest works[3], to correct the errors and evaluate the generated responses.

**Baselines**  In our experiments, we compare our FUSECHAT-7B with four categories of baseline LLMs, including (i) Proprietary LLMs, (ii) Source LLMs, (iii) Ensemble LLMs, and (iv) Fused LLMs. The details of these baselines are shown in Appendix D.

### 4.2 OVERALL RESULTS

In Table 1, we present the overall results of FUSECHAT-7B in comparison with baselines of various architectures and scales on AlpacaEval 2.0 and MT-Bench. Our key observations are as follows. Firstly, after supervised fine-tuning on our high-quality dataset, OpenChat-3.5-7B SFT demonstrates slightly better performance than the pivot LLM OpenChat-3.5-7B. Secondly, in comparison to OpenChat-3.5-7B Multi, which fuses multiple source LLMs simultaneously as FUSELLM (Wan et al., 2024), the target LLMs resulting from pairwise knowledge fusion exhibit superior performance, demonstrating the effectiveness of pairwise fusion in reducing the fusion difficulty. For

---

[3]`https://github.com/lm-sys/FastChat/pull/3158`

| Model | #Params | AlpacaEval 2.0 (GPT-4-1106-Preview) | | MT-Bench (GPT-4-0125-Preview) | | |
|---|---|---|---|---|---|---|
| | | Win Rate | LC Win Rate | 1st Turn | 2nd Turn | Average Score |
| **Proprietary LLMs** | | | | | | |
| GPT-3.5-Turbo-1106(Achiam et al., 2023) | - | 9.18 | 19.30 | 7.56 | 7.41 | 7.48 |
| Claude-3-Opus(Anthropic, 2024) | - | 29.04 | 40.39 | 8.84 | 8.30 | 8.57 |
| GPT-4-1106-Preview(Achiam et al., 2023) | - | 50.00 | 50.00 | 8.86 | 8.71 | 8.79 |
| **Source LLMs** | | | | | | |
| OpenChat-3.5-7B(Wang et al., 2024a) | 7B | 10.20 | 14.90 | 7.14 | 6.55 | 6.84 |
| Starling-LM-7B-alpha(Zhu et al., 2023) | 7B | 14.20 | 14.70 | 7.54 | 6.49 | 7.01 |
| NH2-SOLAR-10.7B(Kim et al., 2023) | 10.7B | 12.22 | 18.13 | 7.11 | 6.36 | 6.74 |
| InternLM2-Chat-20B(Cai et al., 2024) | 20B | 21.70 | 18.70 | 7.78 | 6.34 | 7.06 |
| Mixtral-8x7B-Instruct(Jiang et al., 2024) | 8x7B | 18.30 | 23.70 | 7.76 | 7.00 | 7.38 |
| Qwen1.5-Chat-72B(Bai et al., 2023) | 72B | 26.50 | 36.60 | 7.83 | 7.36 | 7.59 |
| **Ensemble LLMs** | | | | | | |
| Top1-PPL(Mavromatis et al., 2024) | 162B | 25.11 | 27.97 | 7.79 | 6.95 | 7.37 |
| Top1-LLM-Blender(Jiang et al., 2023) | 162B | 24.45 | 29.11 | 7.85 | 6.70 | 7.28 |
| Top1-GPT4(Achiam et al., 2023) | 162B | 42.82 | 43.87 | 8.79 | 8.01 | 8.40 |
| **Fused LLMs** | | | | | | |
| OpenChat-3.5-7B SFT | 7B | 10.56 | 14.50 | 7.36 | 6.40 | 6.88 |
| OpenChat-3.5-7B Multi | 7B | 10.19 (-3.50%) | 13.43 (-7.38%) | 7.69 (+4.48%) | 6.26 (-2.19%) | 6.99 (+1.60%) |
| OpenChat-3.5-7B Starling | 7B | 11.43 (+8.24%) | 16.20 (+11.72%) | 7.69 (+4.48%) | 6.73 (+5.16%) | 7.22 (+4.94%) |
| OpenChat-3.5-7B SOLAR | 7B | 11.12 (+5.30%) | 16.51 (+13.86%) | 7.58 (2.99%) | 6.76 (+5.62%) | 7.17 (+4.22%) |
| OpenChat-3.5-7B InternLM | 7B | **11.82 (+11.93%)** | 15.21 (+4.90%) | 7.63 (+3.67%) | 6.78 (+5.94%) | 7.21 (+4.80%) |
| OpenChat-3.5-7B Mixtral | 7B | 11.74 (+11.17%) | 16.52 (+13.93%) | 7.58 (2.99%) | 6.90 (+7.81%) | 7.24 (+5.23%) |
| OpenChat-3.5-7B Qwen | 7B | 10.93 (+3.50%) | 14.98 (+3.31%) | 7.69 (+4.48%) | 6.78 (+5.94%) | 7.23 (+5.09%) |
| FUSECHAT-7B | 7B | 11.52 (+9.09%) | **17.16 (+18.34%)** | 7.70 (+4.62%) | **7.05 (+10.16%)** | **7.38 (+7.27%)** |

Table 1: Results of FUSECHAT-7B and baselines on AlpacaEval 2.0 and MT-Bench. The bold font denotes the best performance among fused LLMs, while the underscore indicates the second-best performance. Moreover, the percentages represent the relative performance improvement compared to the OpenChat-3.5-7B SFT.

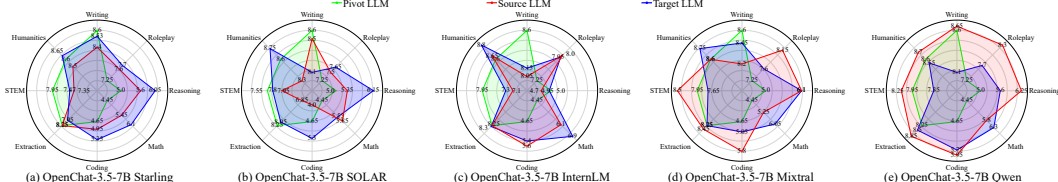

Figure 3: The effect of pairwise knowledge fusion for source LLMs across various domains on MT-Bench. It combines the strengths of each source LLM and the pivot (OpenChat-3.5-7B) into a more potent target LLM.

instance, through the integration of OpenChat-3.5-7B and Mixtral-8x7B-Instruct, the fused target LLM OpenChat-3.5-7B Mixtral achieves relative gains of 13.93% LC Win Rate and 5.23% Average Score over OpenChat-3.5-7B SFT, significantly surpassing OpenChat-3.5-7B Multi. Furthermore, after merging these target LLMs, FUSECHAT-7B shows substantial performance enhancements of 18.34% and 7.27% in the two metrics. This illustrates the superiority of FUSECHAT-7B across a range of source LLMs at various scales, even comparable to 8x7B MoEs and approaching GPT-3.5.

Moreover, in comparison to the ensemble LLMs of 162B, which generate the 1st response from six parallel deployed LLMs based on different ranking criteria, FUSECHAT-7B outperforms most of these LLMs except Top1-GPT4 on MT-Bench, while being 23x smaller and independent of GPT-4.

To further illustrate that our performance improvements stem from the integration of distinct knowledge from multiple LLMs, we evaluate the source LLMs, target LLMs, and FUSECHAT across various domains on MT-Bench. The results in Figure 3 reveal that the target LLMs demonstrate noticeable performance enhancements in most domains after pairwise knowledge fusion. Typically, the performance of each target LLM falls between that of the pivot LLM and the respective source LLM. This phenomenon can be attributed to the fusion function we employed to select the optimal target distributions with minimal cross-entropy, which promotes the incorporation of unique advantages from the pivot LLM and source LLMs into more potent target LLMs. Notably, in math and coding domains, the performance of certain target LLMs surpasses that of either the pivot or source

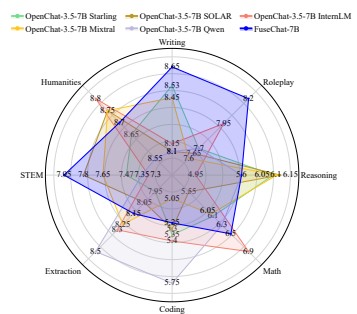

Figure 4: The impact of merging target LLMs into FUSECHAT-7B across domains on MT-Bench.

LLMs. This enhancement can be explained by the strong performance of the source LLMs in these domains, coupled with the relatively high proportion of math and coding samples in our dataset. It is also consistent with findings from knowledge distillation (Wu et al., 2023), where the student model occasionally outperforms the teacher in specific tasks. The effect of further merging these target LLMs into FUSECHAT-7B is shown in Figure 4. By integrating the capabilities of the target LLMs, FUSECHAT achieves a balanced and robust performance across diverse domains.

## 4.3 DIFFERENT PIVOT LLM

We conduct experiments using Starling-LM-7B-alpha to replace OpenChat-3.5-7B as a more robust pivot LLM, which achieved an LC Win Rate of 14.70 on AlpacaEval 2.0 and an Average Score of 7.01 on MT-Bench. The evaluation results presented in Table 2 show that FUSECHAT-Starling-7B outperforms Starling-LM-7B-alpha, with relative performance improvements of 17.62% on AlpacaEval 2.0 and 2.14% on MT-Bench. Notably, although Starling-LM-7B-alpha SFT does not result in performance gains, the pairwise knowledge fusion and model merging processes lead to significant enhancements using the same training data.

| Model | AlpacaEval 2.0 | MT-Bench |
|---|---|---|
| Starling-LM-7B-alpha | 14.70 | 7.01 |
| Starling-LM-7B-alpha SFT | 13.20 (-10.20%) | 6.89 (-1.71%) |
| FUSECHAT-Starling-7B | 17.29 (+17.62%) | 7.16 (+2.14%) |

Table 2: Starling-LM-7B-alpha as pivot LLM results on AlpacaEval 2.0 and MT-Bench.

## 4.4 DATASET SCALING

We perform experiments across different dataset scales for pairwise knowledge fusion, followed by merging the resulting target LLMs in the parameter space to obtain the final fused LLM. The results in Table 3 indicate that the performance of the final fused LLM consistently improves as the training data scales up from 10k to 95k on MT-Bench, demonstrating the potential effectiveness of scaling up dataset to our method.

| Dataset Scale | MT-Bench | | |
|---|---|---|---|
| | 1st Turn | 2nd Turn | Average Score |
| 10,000 | 7.34 | 6.86 | 7.10 |
| 25,000 | 7.58 | 6.85 | 7.21 |
| 95,000 | 7.70 | 7.05 | 7.38 |

Table 3: Comparison of different dataset scales on MT-Bench.

## 4.5 ANALYSIS OF MODEL MERGING

To investigate the effectiveness of the proposed SCE approach, we incorporate the target LLMs using different merging methods, including Linear (Wortsman et al., 2022), TA (Ilharco et al., 2023), TIES (Yadav et al., 2024), and DARE (Yu et al., 2023). We evaluate the performance of these merged LLMs on AlpacaEval 2.0 and MT-Bench. As depicted in Table 4, FUSECHAT-7B SCE outperforms all baseline merging methods on the two benchmarks. For more details of baselines and hyperparameters in model merging, please refer to Appendix B.

| Model | AlpacaEval 2.0 | MT-Bench |
|---|---|---|
| FUSECHAT-7B Linear | 17.12 | 7.03 |
| FUSECHAT-7B TA | 15.74 | 7.08 |
| FUSECHAT-7B TIES | 16.55 | 7.33 |
| FUSECHAT-7B DARE | 16.57 | 7.15 |
| FUSECHAT-7B SCE | 17.16 | 7.38 |

Table 4: Comparison of different merging methods on AlpacaEval 2.0 and MT-Bench.

In Figure 5, we further illustrate the performance of FUSECHAT-7B SCE by incorporating varying numbers of target LLMs on MT-Bench. The findings demonstrate a progressive enhancement in Average Score, which increases from 7.24 to 7.38 as the number of integrated target LLMs rises from 1 to 5. Moreover, we observe that after the integration of OpenChat-3.5-7B SOLAR, the performance of the merged LLM remains stable. This stabilization might be attributed to the comparatively suboptimal performance of OpenChat-3.5-7B SOLAR and its corresponding NH2-SOLAR-10.7B compared to other target or source LLMs. Therefore, we suggest that both the diversity and quality of integrated source LLMs are critical factors for optimal knowledge fusion.

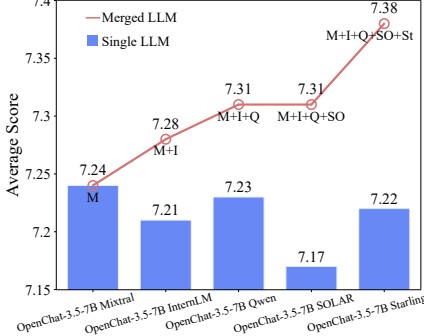

Figure 5: Results of FUSECHAT by merging varying numbers of target LLMs.

## 4.6 ABLATION STUDIES FOR SCE

In this section, we conduct experiments to examine the effectiveness of the *select*, *calculate*, and *erase* operations in SCE. The results in Table 5 demonstrate that, without the *select* operations, FUSECHAT-7B CE suffers substantial performance degradation on the two benchmarks. This underscores the benefits of selecting salient elements from fusion vectors with high variance among target LLMs to signify their distinctive and significant strengths. Moreover, removing both the *select* and *erase* operations leads

| Model | AlpacaEval 2.0 | MT-Bench |
|---|---|---|
| FUSECHAT-7B SCE | 17.16 | 7.38 |
| FUSECHAT-7B CE | 15.69 (-8.57%) | 7.29 (-1.22%) |
| FUSECHAT-7B C | 16.62 (-3.15%) | 7.11 (-3.66%) |

Table 5: Comparison of different merging methods on AlpacaEval 2.0 and MT-Bench. "CE" and "C" mean only the *calculate&erase* and *calculate* operations are used.

to FUSECHAT-7B C with decreased performance on MT-Bench, highlighting the importance of resolving parameter interference in fusion vectors from different target LLMs.

## 4.7 ANALYSIS OF TOKEN ALIGNMENT

Finally, we delve into exploring the impact of various token alignment strategies. Specifically, we apply EM (Fu et al., 2023) and MinED (Wan et al., 2024), and our MS methods to align distributions generated by InterLM2-Chat-20B with those of OpenChat-3.5-7B. Then, we conduct pairwise knowledge fusion to derive OpenChat-3.5-7B InternLM. As depicted in Figure 6, our proposed MS method, rooted in mapping statistics, consistently outperforms EM and MinED, which rely on exact matching and minimal edit distance, respectively. We propose that this performance enhancement arises from MS's effective utilization of token mapping statistics within the dataset, which greatly improves the effect of token alignment in the distribution dimension.

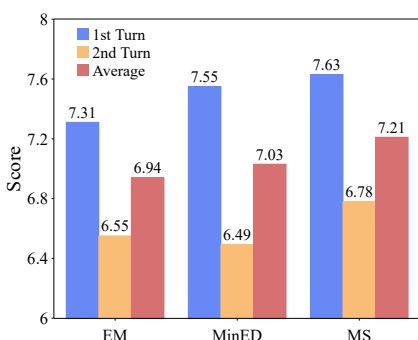

Figure 6: Results of OpenChat-3.5-7B InternLM via pairwise knowledge fusion with different token alignment strategies.

## 5 CONCLUSION

In this work, we propose a fuse-and-merge framework for knowledge fusion of structurally and scale-varied chat LLMs to integrate their collective knowledge and individual strengths into a more potent chat LLM, resulting in FUSECHAT. FUSECHAT first undertakes pairwise knowledge fusion for source chat LLMs to derive multiple target LLMs of identical structure and size via lightweight fine-tuning. Then, these target LLMs are merged within the parameter space using a novel method SCE to calculate the merging coefficients based on the magnitude of parameter updates before and after fine-tuning. Experimental results on two representative instruction-following benchmarks demonstrate the superiority of FUSECHAT across different model scales, even comparable to Mixtral-8x7B-Instruct and approaching GPT-3.5-Turbo-1106 on MT-Bench.

The concept of knowledge fusion shares similarities with related approaches, such as the recently popular mixture of experts (MoEs). Both methods aim to leverage the strengths of multiple models (experts). However, while MoEs require loading multiple experts during inference, leading to higher time and memory requirements, knowledge fusion allows the integration of multiple LLMs with diverse architectures and scales into a single LLM without additional time or memory overhead. This makes knowledge fusion more efficient, especially when model size is a critical consideration.

## 6 LIMITATIONS AND FUTURE WORK

Our work relies on constructing a knowledge fusion dataset that spans diverse domains and leverages the strengths of source LLMs. This process demands substantial data engineering efforts, which may limit the scalability of our methodology. Future research should focus on developing more efficient data synthesis techniques to expand the scope of the knowledge fusion dataset. Additionally, while our study shows improvements in chat model capabilities, it does not address other critical aspects of LLMs, such as knowledge comprehension and the mitigation of hallucinations. Further investigation is necessary to evaluate the applicability and effectiveness of our approach in these areas.

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

# A  DETAILS OF TOKEN ALIGNMENT

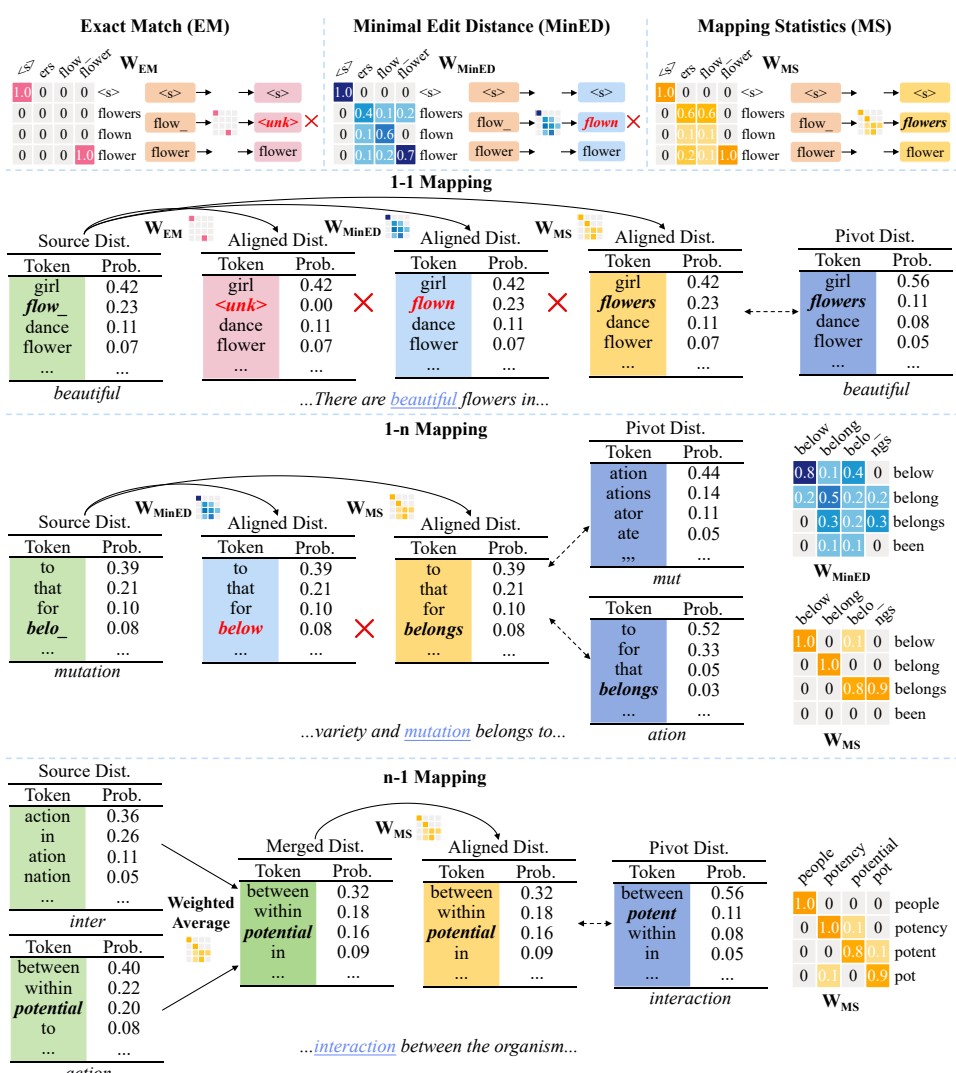

Figure 7: Illustration of EM, MinED, and our MS token alignment strategies in 1-1, 1-n, and n-1 mappings.

In Figure 7, we present the token pair mappings employed in three distinct token alignment strategies, including EM (Fu et al., 2023), MinED (Wan et al., 2024), and our MS. For clarity, these mapping strategies are depicted in a matrix format, where each column represents the probability of a source token being aligned with a corresponding pivot token. The values within these matrices derive from the respective alignment strategies employed. For instance, the matrix $\mathbf{W_{EM}}$ relies on exact matches between source and pivot token pairs, while $\mathbf{W_{MinED}}$ inversely relates to the edit distance between these pairs. $\mathbf{W_{MS}}$ is based on the statistical mapping frequency between the source and pivot tokens.

In scenarios involving 1-1 or 1-n mappings, the EM and MinED methods utilize $\mathbf{W_{EM}}$ or $\mathbf{W_{MinED}}$ respectively, which may lead to inaccurate mappings. For example, in EM, the token "flow_" might be incorrectly aligned with "<unk>", and in MinED, "flow_" could map to "flown", or "belo_" to "below". In contrast, our MS method achieves more accurate alignments such as mapping "flow_" to "flowers" and "belo_" to "belongs", using $\mathbf{W_{MS}}$ from sequence-dimensional token alignments. For n-1 mapping, where only MS is applicable, multiple source distributions are combined using a weighted average determined by $\mathbf{W_{MS}}$ to derive a merged distribution. This unified distribution is then processed similarly to the 1-1 mappings.

## B DETAILS OF MODEL MERGING

The hyperparameters for various merging methods are detailed as follows. For the Linear method (Wortsman et al., 2022), merging parameters are calculated as the mean of all target LLMs. In the TA method (Ilharco et al., 2023), we adhere to the original paper, exploring scaling coefficients within the range of [0.3, 0.4, 0.5]. The optimal setting of 0.3 is selected based on performance. For the TIES (Yadav et al., 2024) and DARE (Yu et al., 2023) approaches, we search for the trim/drop rate within the range of [0.1, 0.2, $\cdots$, 0.9]. The optimal trim/drop rate is 0.4, which results in the elimination of the bottom/random 40% of delta parameters by resetting them to zero. Merging coefficients are computed as the average of all target LLMs. For the SCE method, we search for the salient element selection thresholds $\tau$ within the range of [10, 20, $\cdots$, 90]. The optimal threshold is 10%. Merging coefficients are automatically derived based on the magnitude of delta parameters.

## C DETAILS OF TRAINING DATASET

We curated a comprehensive training dataset, FUSECHAT-MIXTURE, from various sources. This dataset covers different styles and capabilities, featuring both human-written and model-generated, and spanning general instruction-following and specific skills. These sources include:

**Orca-Best**[4]: We sampled 20,000 examples from Orca-Best, which is filtered from the GPT-4 (1M) partition of Orca (Mukherjee et al., 2023) based on maximum length and clustering of instructions.

**Capybara**[5]: We incorporated all the 16,000 examples of Capybara, which is a high-quality collection of multi-turn synthetic conversations.

**No-Robots**[6]: We included all the 9,500 examples of No-Robots, which is a dataset created by skilled human annotators for supervised fine-tuning.

**ShareGPT-GPT4**[7]: We utilized all 6,200 examples from ShareGPT-GPT4, which exclusively uses dialogues generated by GPT-4 in ShareGPT.

**Oasst-Top1**[8]: We selected 5,000 examples from Oasst-Top1, which is a refined version of Oasst1 (Köpf et al., 2024), a human-annotated assistant-style conversation dataset.

**MetaMathQA** [9]: We sampled 10,000 examples from MetaMathQA (Yu et al., 2024), which is augmented from the GSM8K (Cobbe et al., 2021) and MATH (Hendrycks et al., 2021) datasets for mathematics problem-solving.

**OSS-Instruct** [10]: We chose 10,000 examples from OSS-Instruct (Wei et al., 2023), which contains code instruction data synthesized from open-source code snippets.

**Evol-Alpaca** [11]: We sampled 10,000 examples from Evol-Alpaca, which is a code instruction dataset generated by GPT-4 with evol-instruct proposed by WizardCoder (Luo et al., 2024).

**Python-Code** [12]: We selected 10,000 examples from Python-Code, which comprises instructions and responses generated by GPT-3.5 and GPT-4 for python code generation.

We followed the data processing code in FastChat (Zheng et al., 2024) to clean instances containing non-English or special characters. Then, we split long conversations into blocks with a maximum length of 2048 tokens, resulting in the final FUSECHAT-MIXTURE with 95,000 samples. We also explored the domain distribution of the samples in the training data. Specifically, we used the approach provided by Magpie (Xu et al., 2024), utilizing the Llama-3-8B-Instruct model (Dubey et al., 2024) to classify our 95,000 training examples into eight distinct domains as defined by

---

[4]`https://huggingface.co/datasets/shahules786/orca-best`
[5]`https://huggingface.co/datasets/LDJnr/Capybara`
[6]`https://huggingface.co/datasets/HuggingFaceH4/no_robots`
[7]`https://huggingface.co/datasets/shibing624/sharegpt_gpt4`
[8]`https://huggingface.co/datasets/OpenAssistant/oasst_top1_2023-08-25`
[9]`https://huggingface.co/datasets/meta-math/MetaMathQA`
[10]`https://huggingface.co/datasets/ise-uiuc/Magicoder-OSS-Instruct-75K`
[11]`https://huggingface.co/datasets/theblackcat102/evol-codealpaca-v1`
[12]`https://huggingface.co/datasets/ajibawa-2023/Python-Code-23k-ShareGPT`

MT-Bench. After removing 7,000 samples due to anomalous classification errors, the final domain distribution is presented in Table 6, which demonstrates substantial diversity, which aligns with our primary objective to assess the model's general capabilities rather than domain-specific performance.

| Statistics | Math | Extraction | Roleplay | Writing | STEM | Reasoning | Humanities | Coding | Total |
|---|---|---|---|---|---|---|---|---|---|
| Num. Sample | 15079 | 20329 | 8137 | 7627 | 983 | 7948 | 1403 | 27119 | 88625 |
| Percentage (%) | 17.01 | 22.94 | 9.18 | 8.61 | 1.11 | 8.97 | 1.58 | 30.60 | 100 |

Table 6: The domain distribution of samples in the training dataset.

# D  DETAILS OF BASELINES

In this section, we present the details of baseline models compared in our experiments.

**Proprietary LLMs**: GPT-3.5-Turbo-1106[13] (Achiam et al., 2023), Claude-3-Opus[14] (Anthropic, 2024), and GPT-4-1106-Preview[15] (Achiam et al., 2023).

**Source LLMs**: OpenChat-3.5-7B[16] (Wang et al., 2024a), Starling-LM-7B-alpha[17] (Zhu et al., 2023), NH2-SOLAR-10.7B[18] (Kim et al., 2023), InternLM2-Chat-20B[19] (Cai et al., 2024), Mixtral-8x7B-Instruct[20] (Jiang et al., 2024), and Qwen-1.5-Chat-72B[21] (Bai et al., 2023).

**Ensemble LLMs**: Top1-PPL (Mavromatis et al., 2024), which selects the 1st ranked response from source LLMs based on the perplexity of instruction; Top1-LLM-Blender (Jiang et al., 2023), which ranks and combines the output text from source LLMs with ranker and fuser models. Due to the fuser model's constraints on maximum sequence length, only the ranker model is utilized to determine and produce the 1st-ranked response; Top1-GPT4 (Achiam et al., 2023), which leverages GPT-4 judgment as ranking criteria and yields the 1st ranked response. Since our evaluations also rely on GPT-4, this approach represents an upper bound for comparison.

**Fused LLMs**: OpenChat-3.5-7B SFT, a special scenario of knowledge fusion with a single source LLM, serves as the supervised fine-tuning baseline using our training dataset; OpenChat-3.5-7B Multi is the knowledge fusion of multiple source chat LLMs simultaneously like FUSELLM (Wan et al., 2024); OpenChat-3.5-7B Starling, OpenChat-3.5-7B SOLAR, OpenChat-3.5-7B InternLM, OpenChat-3.5-7B Mixtral, and OpenChat-3.5-7B Qwen are target LLMs resulting from pairwise knowledge fusion of the pivot LLM OpenChat-3.5-7B and the rest source LLMs.

# E  EVALUATION OF ADDITIONAL BENCHMARKS

The primary objective of FUSECHAT is to integrate multiple chat LLMs into a more powerful model. Consequently, our experiments primarily focus on alignment training data, such as ShareGPT, and chat model evaluation benchmarks like AlpacaEval 2.0 and MT-Bench. In addition to the chat model benchmarks, we also conducted experiments on six general evaluation benchmarks, including MMLU-Pro (Wang et al., 2024b), PIQA (Bisk et al., 2020), BoolQ (Clark et al., 2019), GPQA (Rein et al., 2023), GSM8K (Cobbe et al., 2021), and IFEval (Zhou et al., 2023), which assess knowledge understanding, question-answering, common-sense reasoning, and instruction-following. The results are presented in Table 7. It is important to note that the training data for FUSECHAT-7B is primarily focused on alignment rather than general knowledge. Therefore, performance improvements on these general benchmarks are less significant compared to those on AlpacaEval 2.0 and MT-Bench. This

---

[13] https://platform.openai.com/docs/models/gpt-3-5-turbo
[14] https://www.anthropic.com/news/claude-3-family
[15] https://platform.openai.com/docs/models/gpt-4-turbo-and-gpt-4
[16] https://huggingface.co/openchat/openchat_3.5
[17] https://huggingface.co/berkeley-nest/Starling-LM-7B-alpha
[18] https://huggingface.co/NousResearch/Nous-Hermes-2-SOLAR-10.7B
[19] https://huggingface.co/internlm/internlm2-chat-20b
[20] https://huggingface.co/mistralai/Mixtral-8x7B-Instruct-v0.1
[21] https://huggingface.co/Qwen/Qwen1.5-72B-Chat

observation is consistent with recent studies on alignment (Meng et al., 2024; Wu et al., 2024), which highlight the critical role of alignment dataset construction in determining downstream performance.

| Model | MMLU-Pro | PIQA | BoolQ | GPQA | GSM8K | IFEval | Average |
|---|---|---|---|---|---|---|---|
| OpenChat-3.5-7B | 31.63 | 82.86 | 73.91 | 31.30 | 76.88 | 35.73 | 55.38 |
| OpenChat-3.5-7B SFT | 31.32 | 82.75 | 73.91 | 30.30 | 76.04 | 35.34 | 54.94 |
| OpenChat-3.5-7B Multi | 31.39 | 82.43 | 73.73 | 32.30 | 74.75 | 36.25 | 55.14 |
| FUSECHAT-7B | 31.65(+0.06%) | 82.97(+0.13%) | 75.50(+2.15%) | 37.40(+19.49%) | 77.10(+0.29%) | 37.49(+4.93%) | 57.02(+2.96%) |

Table 7: Comparison results on general evaluation benchmarks.

# F STATISTICAL SIGNIFICANCE OF PERFORMANCE IMPROVEMENTS

We conduct detailed statistical analyses using t-tests to evaluate the performance of our proposed FUSECHAT-7B model on MT-Bench compared to two key baselines: Pairwise Fusion and OpenChat-3.5-7B Multi, which fuses multiple source LLMs simultaneously.

The results, summarized in Table 8, highlight the strong statistical significance of the performance improvements achieved by FUSECHAT-7B over these baselines. Notably, the p-values are well below the 0.05 threshold, confirming the significance of the observed differences. These findings provide strong evidence supporting the superiority of FUSECHAT-7B compared to both Pairwise Fusion and OpenChat-3.5-7B Multi.

| Model Comparison | t-statistic | p-value |
|---|---|---|
| FUSECHAT-7B vs. Pairwise Fusion | 2.95874 | 0.00318 |
| FUSECHAT-7B vs. OpenChat-3.5-7B Multi | 3.32756 | 0.00108 |

Table 8: T-test results comparing FUSECHAT-7B with Pairwise Fusion and OpenChat-3.5-7B Multi on MT-Bench, highlighting the statistical significance of performance improvements across the models.

# G COMPARISON OF PAIRWISE FUSION AND SINGLE-MODEL DISTILLATION

The key distinction between pairwise fusion and single-model distillation lies in their learning paradigms. In pairwise fusion, the model selectively acquires knowledge based on the quality of outputs from the source LLM or pivot LLM, guided by lower cross-entropy (CE) values. This approach ensures that the model consistently learns from the stronger performer for each sample. In contrast, single-model distillation relies exclusively on the source LLM, implicitly assuming that the source consistently provides superior outputs.

To rigorously assess the differences between pairwise fusion and single-model distillation, we conducted additional experiments. Specifically, the pairwise fusion strategy in FUSECHAT was replaced with direct distillation from a single source model, omitting the merging phase. The results, summarized in Table 9, demonstrate that pairwise fusion consistently outperforms single-model distillation across five source LLMs. For clarity, the notation D/P indicates the perfor-

| Model | AlpacaEval 2.0 | MT-Bench |
|---|---|---|
| OpenChat-3.5-7B Qwen (D/P) | 5.98/**14.98** | 6.79/**7.23** |
| OpenChat-3.5-7B Mixtral (D/P) | 16.10/**16.52** | 7.03/**7.24** |
| OpenChat-3.5-7B InternLM (D/P) | 6.54/**15.21** | 6.88/**7.21** |
| OpenChat-3.5-7B SOLAR (D/P) | 12.21/**16.51** | 7.09/**7.17** |
| OpenChat-3.5-7B Starling (D/P) | 14.89/**16.20** | 7.15/**7.22** |

Table 9: Comparison of pairwise fusion (P) and single-model distillation (D) across five source LLMs, evaluated on AlpacaEval-2.0 and MT-Bench.

mance of direct distillation and pairwise fusion, respectively. The metrics reported include the Average Score on MT-Bench and the Length-Controlled Win Rate on AlpacaEval 2.0.

Furthermore, the SCE method was applied to fuse the models obtained through single-model distillation. As shown in Table 10, the results reveal that merging the models derived from pairwise fusion produces a superior fused model compared to merging models from single-model distillation. These results highlight the effectiveness of the pairwise fusion approach, not only in standalone performance but also in enhancing the quality of the final fused model.

| Model | AlpacaEval 2.0 | MT-Bench |
|---|---|---|
| FUSECHAT-7B (D/P) | 14.68/**17.16** | 6.91/**7.38** |

Table 10: Performance comparison of the final FUSECHAT-7B models derived from single-model distillation (D) and pairwise fusion (P) using the SCE method.

## H  RATIONALE BEHIND THE VALUE OF $\lambda$

In Eq. 4, $\lambda$ is set to 0.9 to balance the contributions of the SFT loss and the fusion loss. This value is carefully chosen due to the substantial difference in magnitude between these two losses. To illustrate this, we conducted an experiment using Qwen-1.5-Chat-72B as the source LLM and randomly selected 128 instances from the training dataset.

The observed loss values are presented in Table 11. The results indicate that the fusion loss is approximately three times larger than the SFT loss in this setting. This disparity necessitates assigning a proportionally smaller weight to the fusion loss to prevent it from dominating the optimization process. Without this adjustment, an overly high fusion loss weight could distort the training dynamics, leading to suboptimal learning. Therefore, the 0.9/0.1 weight distribution is a carefully calibrated choice designed to ensure balanced optimization and effective learning within the framework of Eq. 4.

| Loss Type | Loss Value |
|-----------|------------|
| SFT       | 0.5077     |
| Fusion    | 1.3081     |

Table 11: Loss values for SFT and fusion during training with Qwen-1.5-Chat-72B as the source LLM.

## I  CASE STUDIES

We present case studies to demonstrate the individual strengths of target LLMs (OpenChat-3.5-7B Starling, OpenChat-3.5-7B SOLAR, OpenChat-3.5-7B InternLM, OpenChat-3.5-7B Mixtral, and OpenChat-3.5-7B Qwen) obtained from pairwise knowledge fusion of source LLMs, and show the collective knowledge and strengths of FUSECHAT (FUSECHAT-7B) obtained by further merging target LLMs. OpenChat-3.5-7B SFT is used as the baseline for comparison.

---

**OpenChat-3.5-7B SFT; Math; ID 113;**

**Turn 1 Question:** In a survey conducted at a local high school, preferences for a new school color were measured: 58% of students liked the color blue, 45% preferred green, and 22% liked both colors. If we randomly pick a student from the school, what's the probability that they would like neither blue nor green?

**Turn 1 Answer:** Let's use the principle of inclusion-exclusion to solve this problem. The total percentage of students who like either blue or green is the sum of the individual percentages minus the percentage who like both:

$$58\%(blue) + 45\%(green) - 22\%(both) = 75\%$$

This means that 100% - 75% = 25% of students like neither blue nor green. So, the probability that a randomly picked student would not like neither blue nor green is $\boxed{25\%}$. The answer is: 25

**Turn 1 Judgment:** The assistant's answer contains a mathematical error in calculating the total percentage of students who like either blue or green. The correct calculation, as shown in the reference answer, should be:

$$58\% + 45\% - 22\% = 81\%$$

This means that 81% of students like at least one of the colors, not 75% as the assistant stated. Consequently, the correct percentage of students who like neither color is:

$$100\% - 81\% = 19\%$$

Therefore, the correct probability that a randomly picked student would like neither blue nor green is 19%, not 25% as the assistant incorrectly concluded. The assistant's mistake significantly affects the correctness of the response. However, the assistant did correctly apply the principle of inclusion-exclusion, which indicates an understanding of the correct approach to solving the problem. The error seems to be purely in the arithmetic calculation. Given the importance of accuracy in mathematical problems and the impact of this error on the final answer, the response's helpfulness is diminished. However, the attempt to apply the correct principle shows some level of understanding. Rating: [[3]]

**Turn 2 Question:** If we select a student liked green, what's the probability that he or she would dislike both colors?

**Turn 2 Answer:** Since 22% of students like both colors, this means that 78% of students do not like both colors. Out of these, 45% prefer green. Therefore, the probability that a student who likes green would dislike the other color is $45\% \times 78\% = 35.7\%$. So, the probability that a student who likes green would dislike the other color is $\boxed{35.7\%}$. The answer is: 35.7

**Turn 2 Judgment:** The assistant's response to the second question contains a fundamental misunderstanding of the question posed. The question asks for the probability that a student who likes green would dislike both colors, which is a contradiction. If a student likes green, they cannot simultaneously dislike green, making it impossible for them to dislike both green and another color (in this case, blue). The correct answer, as provided in the reference answer, is 0%. This is because the premise of selecting a student who likes green excludes the possibility of that student disliking both colors. The assistant's calculation of 35.7% is incorrect because it misinterprets the question as asking for the probability that a student who likes green would dislike the other color (blue), which was not the question asked. Therefore, the assistant's response is incorrect due to a misunderstanding of the question's premise, leading to an erroneous calculation that does not apply to the scenario described. Rating: [[1]]

---

Table 12: Case studies on MT-Bench. The answers are generated by OpenChat-3.5-7B SFT.

| |
|---|
| **OpenChat-3.5-7B Starling; Math; ID 113;** |
| **Turn 1 Question:** In a survey conducted at a local high school, preferences for a new school color were measured: 58% of students liked the color blue, 45% preferred green, and 22% liked both colors. If we randomly pick a student from the school, what's the probability that they would like neither blue nor green? |
| **Turn 1 Answer:** We can solve this problem using the principle of inclusion-exclusion. The total number of students who like either blue or green is the sum of those who like blue, those who like green, minus those who like both (to avoid double-counting). So, the number of students who like either color is: $$58\%(blue) + 45\%(green) - 22\%(both) = 75\%$$ This means that 25% of students like neither color. Therefore, the probability that a randomly picked student likes neither color is $\boxed{25\%}$. The answer is: 25 |
| **Turn 1 Judgment:** The assistant's answer contains a mathematical error in calculating the percentage of students who like either blue or green. The correct calculation, as shown in the reference answer, should be: $$\text{At least one of the colors} = 58\% + 45\% - 22\% = 103\% - 22\% = 81\%$$ This means that 81% of students like at least one of the colors, not 75% as the assistant stated. Consequently, the percentage of students who like neither color is not 25% but rather: $$\text{Neither color} = 100\% - 81\% = 19\%$$ Therefore, the correct probability that a randomly picked student likes neither color is 19%, not 25%. The assistant's mistake significantly affects the correctness of the response. The correct approach and calculation were provided in the reference answer, which the assistant failed to replicate accurately. This error directly impacts the helpfulness of the response, as it provides the user with incorrect information. Given the importance of accuracy in mathematical problems and the potential confusion caused by incorrect percentages, the assistant's response is rated lower due to the critical error in calculation. Rating: [[3]] |
| **Turn 2 Question:** If we select a student liked green, what's the probability that he or she would dislike both colors? |
| **Turn 2 Answer:** Since the student dislikes both colors, they must dislike green. Therefore, the probability that a student who likes green would dislike both colors is 0%. |
| **Turn 2 Judgment:** The assistant's response to the second question is correct and aligns with the reference answer. It correctly states that if a student likes green, the probability that they would dislike both colors (blue and green) is 0%. This is because the premise of the question already establishes that the student likes green, making it impossible for them to dislike both colors simultaneously. The explanation is clear, concise, and logically sound, directly addressing the user's question without unnecessary complexity. The assistant's answer to the second question does not contain any mistakes and is both correct and helpful. It provides a straightforward explanation that directly answers the user's query, making it easy for the user to understand the concept being discussed. Rating: [[10]] |

Table 13: Case studies on MT-Bench. The answers are generated by OpenChat-3.5-7B Starling.

**OpenChat-3.5-7B SOLAR; Math; ID 113;**

**Turn 1 Question:** In a survey conducted at a local high school, preferences for a new school color were measured: 58% of students liked the color blue, 45% preferred green, and 22% liked both colors. If we randomly pick a student from the school, what's the probability that they would like neither blue nor green?

**Turn 1 Answer:** We can solve this problem using the principle of inclusion-exclusion. The total number of students who like either blue or green is the sum of those who like blue and those who like green, minus the number who like both. That is:

$$\text{Total} = \text{Blue} + \text{Green} - \text{Both}$$

We are given that 58% of students like blue, 45% like green, and 22% like both. So, we can calculate the percentage of students who like neither as follows:

$$\text{Neither} = \text{Total} - \text{Blue} - \text{Green} + \text{Both}$$

Substituting the given percentages:

$$\text{Neither} = 100\% - (58\% + 45\% - 22\%)$$

Simplifying:

$$\text{Neither} = 100\% - 75\%$$

Therefore, the probability that a randomly picked student would like neither blue nor green is 25%.

**Turn 1 Judgment:** The assistant's answer contains a critical error in the calculation process. The mistake lies in the incorrect application of the principle of inclusion-exclusion and the subsequent calculation of the percentage of students who like neither color. The correct application of the principle of inclusion-exclusion for calculating the percentage of students who like at least one of the colors (blue or green) is as follows:

$$\text{At least one of the colors} = (\text{Blue}) + (\text{Green}) - (\text{Both})$$

$$\text{At least one of the colors} = 58\% + 45\% - 22\% = 103\% - 22\% = 81\%$$

This means 81% of students like at least one of the colors, blue or green. To find the percentage of students who like neither color, we subtract this percentage from 100%:

$$\text{Neither color} = 100\% - \text{At least one of the colors}$$

$$\text{Neither color} = 100\% - 81\% = 19\%$$

Therefore, the correct probability that a randomly picked student would like neither blue nor green is 19%, not 25% as the assistant incorrectly calculated. The assistant's mistake was in the formula used for calculating the percentage of students who like neither color. The assistant incorrectly attempted to subtract the sum of individual preferences minus the overlap from 100%, which does not align with the principle of inclusion-exclusion correctly applied in this context. Given the significant error in the calculation that led to an incorrect final answer, the response's correctness is compromised. However, the assistant did correctly identify the need to use the principle of inclusion-exclusion, which shows an understanding of the correct approach, albeit misapplied. Rating: [[3]]

**Turn 2 Question:** If we select a student liked green, what's the probability that he or she would dislike both colors?

**Turn 2 Answer:** If a student is selected who likes green, the probability that he or she would dislike both colors is 1 - the probability that they like both colors. We know that 22% of students like both colors. So, the probability that a student who likes green dislikes both colors is:

$$1 - 22\% = 78\%$$

Therefore, the probability that a student who likes green would dislike both colors is 78%.

**Turn 2 Judgment:** The assistant's response to the second question contains a fundamental misunderstanding of the scenario presented. According to the reference answer, if a student likes green, the probability that they would dislike both colors is 0%. This is logically consistent because liking green precludes the possibility of disliking both green and blue simultaneously. The assistant's answer, suggesting a 78% probability that a student who likes green would dislike both colors, is incorrect. The correct answer should acknowledge that once a student is known to like green, they cannot be in the category of disliking both colors, making the probability 0%. Rating: [[1]]

Table 14: Case studies on MT-Bench. The answers are generated by OpenChat-3.5-7B SOLAR.

**OpenChat-3.5-7B InternLM; Math; ID 113;**

**Turn 1 Question:** In a survey conducted at a local high school, preferences for a new school color were measured: 58% of students liked the color blue, 45% preferred green, and 22% liked both colors. If we randomly pick a student from the school, what's the probability that they would like neither blue nor green?

**Turn 1 Answer:** To find the probability that a student likes neither blue nor green, we need to subtract the probability that they like at least one of the colors from 1. Let A be the event that a student likes blue, and B be the event that a student likes green. We are given that $P(A) = 0.58$, $P(B) = 0.45$, and $P(A \cap B) = 0.22$. First, we need to find the probability that a student likes at least one of the colors. We can use the formula for the probability of the union of two events:

$$P(A \cup B) = P(A) + P(B) - P(A \cap B)$$

$$P(A \cup B) = 0.58 + 0.45 - 0.22 = 0.75$$

Now, to find the probability that a student likes neither blue nor green, we subtract the probability that they like at least one of the colors from 1:

$$P(neither) = 1 - P(atleastonecolor)$$

$$P(neither) = 1 - 0.75 = 0.25$$

So, the probability that a randomly picked student would like neither blue nor green is 0.25 or 25%.

**Turn 1 Judgment:** The assistant's answer contains a mathematical error in calculating the probability that a student likes at least one of the colors. The assistant incorrectly calculates the probability of a student liking at least one of the colors (blue or green) as 0.75 (or 75%), which leads to an incorrect final answer for the probability of a student liking neither color as 0.25 (or 25%). The correct calculation, as shown in the reference answer, should be as follows:
- The probability of liking blue (P(Blue)) = 58% or 0.58
- The probability of preferring green (P(Green)) = 45% or 0.45
- The probability of liking both colors (P(Both)) = 22% or 0.22
Using the principle of inclusion-exclusion:

$$At least one of the colors = P(Blue) + P(Green) - P(Both) = 0.58 + 0.45 - 0.22 = 0.81$$

This means 81% of students like at least one of the colors, and thus:

$$Neither color = 1 - At least one of the colors = 1 - 0.81 = 0.19$$

Therefore, the correct probability that a randomly picked student would like neither blue nor green is 0.19 or 19%, not 25% as the assistant's answer suggests. The assistant's mistake lies in the incorrect calculation of the union of the two events, which led to an inaccurate conclusion. This error significantly impacts the correctness of the response. Given the importance of accuracy in probability calculations and the impact of this error on the final answer, the assistant's response is rated lower for its correctness. However, the assistant did correctly apply the principle of inclusion-exclusion and provided a clear explanation of the steps, which is helpful for understanding the process. Rating: [[4]]

**Turn 2 Question:** If we select a student liked green, what's the probability that he or she would dislike both colors?

**Turn 2 Answer:** The question seems to be asking for the probability that a student who likes green would dislike both colors, which is not possible because the student already likes green. The student cannot dislike both colors while also liking one of them.

**Turn 2 Judgment:** The assistant's response to the second question is correct and aligns with the reference answer. It correctly identifies the logical inconsistency in the question: if a student likes green, they cannot simultaneously dislike both green and another color. The assistant succinctly explains why the probability that a student who likes green would dislike both colors is 0%, which is the same conclusion reached in the reference answer. This explanation is clear, logically sound, and directly addresses the user's question, demonstrating an understanding of basic probability principles and the specific scenario presented. Rating: [[10]]

Table 15: Case studies on MT-Bench. The answers are generated by OpenChat-3.5-7B InternLM.

| **OpenChat-3.5-7B Mixtral; Math; ID 113;** |
| --- |
| **Turn 1 Question:** In a survey conducted at a local high school, preferences for a new school color were measured: 58% of students liked the color blue, 45% preferred green, and 22% liked both colors. If we randomly pick a student from the school, what's the probability that they would like neither blue nor green? |
| **Turn 1 Answer:** The probability that a student likes blue is 58%, and the probability that they like green is 45%. Since 22% of students like both colors, we can subtract this from the total percentage to find the percentage of students who like either blue or green but not both.

$$58\% + 45\% - 22\% = 71\%$$

So, 71% of students like either blue or green, which means 100% - 71% = 29% of students like neither color.

Therefore, the probability that a randomly picked student would like neither blue nor green is $\boxed{29\%}$. The answer is: 29 |
| **Turn 1 Judgment:** The assistant's answer contains a critical mathematical error in calculating the probability of students liking neither blue nor green. The correct approach, as outlined in the reference answer, involves using the principle of inclusion-exclusion to accurately account for the overlap between students who like both colors. Correct Calculation:
- The correct calculation to find the percentage of students who like at least one of the colors (blue or green) is to add the percentages of students who like each color and then subtract the percentage of students who like both (to correct for double counting), which is $58\% + 45\% - 22\% = 81\%$.
- Therefore, to find the percentage of students who like neither color, we subtract the percentage of students who like at least one of the colors from 100%, resulting in $100\% - 81\% = 19\%$.
The assistant incorrectly calculated the percentage of students who like either blue or green but not both as 71%, leading to the incorrect conclusion that 29% of students like neither color. The correct probability that a randomly picked student would like neither blue nor green is 19%, not 29%. Given the significant mathematical error and the resulting incorrect conclusion, the assistant's answer is not helpful for someone trying to understand how to correctly apply the principle of inclusion-exclusion in probability. Rating: [[2]] |
| **Turn 2 Question:** If we select a student liked green, what's the probability that he or she would dislike both colors? |
| **Turn 2 Answer:** If a student is selected who likes green, the probability that they dislike both colors is equal to the percentage of students who like both colors. In this case, 22% of students like both colors, so the probability that a student who likes green would dislike both colors is $\boxed{22\%}$. The answer is: 22 |
| **Turn 2 Judgment:** The assistant's response to the second question contains a significant error. The correct answer, as outlined in the reference answer, is that the probability of a student who likes green disliking both colors is 0%. This is because the premise of the question already establishes that the student likes green, making it impossible for them to dislike both green and blue simultaneously. The assistant's answer incorrectly states that the probability is 22%, which seems to be a misunderstanding of the question's premise. The correct approach should have recognized the logical contradiction in the question: a student who likes green cannot at the same time dislike both green and another color. Therefore, the probability of a student who likes green also disliking both colors is 0%. Given this significant mistake and misunderstanding of the question's premise, the response is incorrect and misleading. Rating: [[1]] |

Table 16: Case studies on MT-Bench. The answers are generated by OpenChat-3.5-7B Mixtral.

| OpenChat-3.5-7B Qwen; Math; ID 113; |
| --- |
| **Turn 1 Question:** In a survey conducted at a local high school, preferences for a new school color were measured: 58% of students liked the color blue, 45% preferred green, and 22% liked both colors. If we randomly pick a student from the school, what's the probability that they would like neither blue nor green? |
| **Turn 1 Answer:** To find the probability that a student likes neither blue nor green, we can use the formula for the probability of the union of two events:

$$P(A \cup B) = P(A) + P(B) - P(A \cap B)$$

In this case, A represents the event that a student likes blue, and B represents the event that a student likes green. We are given that $P(A) = 0.58$, $P(B) = 0.45$, and $P(A \cap B) = 0.22$. First, we need to find the probability of the event that a student likes either blue or green, which is represented by $P(A \cup B)$:

$$P(A \cup B) = P(A) + P(B) - P(A \cap B) = 0.58 + 0.45 - 0.22 = 0.75$$

Now, we need to find the probability of the event that a student likes neither blue nor green, which is represented by $P(A' \cap B')$:

$$P(A' \cap B') = 1 - P(A \cup B) = 1 - 0.75 = 0.25$$

Therefore, the probability that a randomly picked student would like neither blue nor green is 0.25 or 25%. |
| **Turn 1 Judgment:** The assistant's answer contains a mathematical error in calculating the probability of a student liking either blue or green. The assistant incorrectly calculates the sum of the probabilities as 0.75 (or 75%) instead of correctly calculating it as 81%, as shown in the reference answer. This error leads to an incorrect final answer for the probability of a student liking neither blue nor green. Correct Calculation based on the reference answer:
- The correct calculation for the probability of a student liking at least one of the colors (blue or green) is 58% + 45% - 22% = 81%.
- Therefore, the correct probability of a student liking neither color is 100% - 81% = 19%.
The assistant's answer incorrectly concludes that the probability is 25%, whereas the correct probability, as per the reference answer and the correct calculation, is 19%. Despite the assistant's method being fundamentally sound and the explanation clear, the mathematical mistake significantly impacts the correctness of the final answer. The assistant's approach correctly applies the principle of inclusion-exclusion but fails in the execution of basic arithmetic. Given the importance of accuracy in mathematical problems and the impact of this error on the final answer, the assistant's response must be rated lower. However, the explanation and approach were correctly aligned with the principle of inclusion-exclusion, which is a positive aspect of the response. Rating: [[4]] |
| **Turn 2 Question:** If we select a student liked green, what's the probability that he or she would dislike both colors? |
| **Turn 2 Answer:** If a student is selected who likes green, we want to find the probability that they dislike both colors. Since the student likes green, they cannot dislike both colors. Therefore, the probability that a student who likes green would dislike both colors is 0. |
| **Turn 2 Judgment:** The assistant's response to the second question is correct and aligns with the reference answer. It accurately explains that if a student is already known to like green, it is impossible for them to dislike both green and blue simultaneously. Therefore, the probability that a student who likes green would dislike both colors is indeed 0%. This explanation is clear, concise, and directly addresses the user's question without unnecessary complexity. The assistant's answer is helpful in clarifying the logical impossibility of the scenario proposed by the user, thus preventing any confusion about the nature of probability in this context. Rating: [[10]] |

Table 17: Case studies on MT-Bench. The answers are generated by OpenChat-3.5-7B Qwen.

| FUSECHAT-7B; Math; ID 113; |
| --- |
| **Turn 1 Question:** In a survey conducted at a local high school, preferences for a new school color were measured: 58% of students liked the color blue, 45% preferred green, and 22% liked both colors. If we randomly pick a student from the school, what's the probability that they would like neither blue nor green? |
| **Turn 1 Answer:** To find the probability that a student likes neither blue nor green, we can use the formula: $$P(\text{neither}) = 1 - P(\text{blue}) - P(\text{green}) + P(\text{both})$$ where P(blue) is the probability that a student likes blue, P(green) is the probability that a student likes green, and P(both) is the probability that a student likes both colors. We are given that $P(\text{blue}) = 58\%$, $P(\text{green}) = 45\%$, and $P(\text{both}) = 22\%$. Plugging these values into the formula, we get: $$P(\text{neither}) = 1 - 0.58 - 0.45 + 0.22$$ $$P(\text{neither}) = 1 - 0.85 + 0.22$$ $$P(\text{neither}) = 0.15$$ So, the probability that a randomly picked student would like neither blue nor green is 15%. |
| **Turn 1 Judgment:** The assistant's answer uses the principle of inclusion-exclusion correctly but makes a calculation error in the final step. According to the assistant's formula: $$P(\text{neither}) = 1 - P(\text{blue}) - P(\text{green}) + P(\text{both})$$ The values given are: - $P(\text{blue}) = 58\% = 0.58$ - $P(\text{green}) = 45\% = 0.45$ - $P(\text{both}) = 22\% = 0.22$ Plugging these into the formula, the assistant calculates: $$P(\text{neither}) = 1 - 0.58 - 0.45 + 0.22$$ However, the assistant incorrectly calculates the final result as 0.15 (or 15%). The correct calculation should be: $$P(\text{neither}) = 1 - 0.58 - 0.45 + 0.22 = 1 - 1.03 + 0.22 = 1 - 0.81 = 0.19$$ Therefore, the correct probability that a randomly picked student would like neither blue nor green is 19%, not 15%. The reference answer correctly calculates this probability as 19% using the principle of inclusion-exclusion and correctly accounts for the overlap between students who like both colors. The assistant's approach is fundamentally correct but falters at the arithmetic stage, leading to an incorrect final answer. Given the mistake in calculation, the assistant's answer is not entirely correct, but since the method used was appropriate, it shows a good understanding of the principle involved. Therefore, the assistant's response, while flawed in execution, is still somewhat helpful for understanding how to approach the problem. Rating: [[6]] |
| **Turn 2 Question:** If we select a student liked green, what's the probability that he or she would dislike both colors? |
| **Turn 2 Answer:** If a student is selected who likes green, we want to find the probability that they dislike both colors. Since the student likes green, they cannot dislike both colors. Therefore, the probability that a student who likes green would dislike both colors is 0%. |
| **Turn 2 Judgment:** The assistant's answer to the second question is correct and aligns with the reference answer. It accurately explains that if a student likes green, the probability of them disliking both colors (blue and green) is 0%. This is because the premise of the question already establishes that the student likes green, making it impossible for them to dislike both colors simultaneously. The explanation is clear, concise, and logically sound, effectively addressing the user's query without any errors or unnecessary complexity. Rating: [[10]] |

Table 18: Case studies on MT-Bench. The answers are generated by FUSECHAT-7B.

