# OpenReview forum: "FuseChat: Knowledge Fusion of Chat Models"
_ICLR.cc/2025/Conference — ICLR 2025 Conference Withdrawn Submission_

### Official Review · Reviewer_UpRG · 2024-10-24

**Soundness:** 3
**Presentation:** 2
**Contribution:** 3
**Rating:** 6
**Confidence:** 5

**Summary:**

This paper introduce a fuse-and-merge framework called FUSECHAT, which includes two stages. Pairwise knowledge fusion using a pivot LLM and token alignment to generate target LLMs with identical structure and size, and merging these models via SCE method, which determines merging coefficients based on parameter updates.

**Strengths:**

In general, the logic of the article is good, and the abstract, main text, and conclusions are consistent. The experiments are sufficiently convincing. The author summarizes the previous work from multiple aspects in the related work section.

**Weaknesses:**

1. In the Introduction section, there is insufficient explanation of the challenges faced by FUSECHAT. It is not enough to just explain the advantages of knowledge fusion, but the complexity of the work should also be highlighted.
2. The contribution of the work done in this paper is not explained in the Introduction section.
3. The method section uses too many narrative words and lacks specific formula expressions, which increases the difficulty for readers to understand the article.
4. In the experiment section, there is a lack of explanation for the adverse results in the experiment.

**Questions:**

1. First, the challenges of knowledge fusion tasks and the contributions of this paper should be introduced in the Introduction section.
2. The Method section should highlight the work done by the author. Extensive introduction of work that is not their own will make the article appear less innovative, and you can add formulas to further explain Token Alignment.
3. The introduction of the SCE algorithm is too short, and the reasons for the use of some steps are not introduced, such as the Calculate and Erase steps.
4. Added explanations for poor experimental results in the experimental section, for example, Target LLM performs worse than Pivot LLM and Source LLM in some dimensions.

---

> ### Author Response · Authors · 2024-11-20
> **Official Comment by Authors: Part 1**
>
> Thank you for reviewing our paper and providing insightful feedback. We’re glad you find our work logical and convincing and will address your concerns point by point.
>
> > **Q1: Regarding the challenges of knowledge fusion and FuseChat.**
>
> **A1:** Thank you for your insightful feedback. We appreciate the opportunity to clarify and expand upon this point. In the revised manuscript, we will emphasize the challenges and complexities faced by FuseChat from two key perspectives. First, unlike direct supervised fine-tuning, knowledge fusion introduces additional training costs, such as the computational overhead of inferencing results from the source LLMs. This added step significantly increases resource and time requirements. Second, knowledge fusion methods like FuseChat encounter inherent challenges, including vocabulary alignment across different LLMs and the merging of their distribution matrices. These processes are non-trivial and can introduce noise and errors, which in turn may impact the quality of the fusion results. By addressing these aspects, we aim to provide a more comprehensive discussion of the intricacies involved in developing FuseChat.
>
> > **Q2: Regarding the details and formulas for our token alignment method.**
>
> **A2:** Token alignment is designed to address the mapping challenges between probabilistic distribution matrices generated by different source LLMs for a given instruction. This alignment occurs along two critical dimensions: sequence and probability distribution.
> For sequence alignment, we employ dynamic programming to effectively map the tokenized sequences from the source LLM to those of the pivot LLM. For distribution alignment, we propose to utilize mapping statistics (MS) from the sequence dimension as the criteria for alignment in the distribution dimension. To enhance clarity, we have included a visual illustration of our token alignment method in Figure 7 (Appendix A). We will further refine this explanation by introducing explicit mathematical formulations in the revised revision.
>
> > **Q3: Regarding the motivation and details of the SCE algorithm.**
>
> **A3:** The motivation of our SCE is to design a simple merging strategy to **automatically identify and incorporate the learned advantages from diverse target LLMs while simultaneously resolving knowledge conflicts in the parameter space, without the need for additional parameter tuning**.
>
> To achieve this, we utilize weight updates from the pivot LLM to various target LLMs during the model fusion process, employing these updates as fusion vectors that reflect diverse advantages from different models. The weight merging for each parameter matrix unit in the target LLMs is carried out through a three-step procedure:
>
> 1. Fusion vectors for each unit parameter matrix, derived from various target LLMs, are intended to capture distinctive and significant strengths of these models. To emphasize the most impactful features, we select the top τ% of elements from each parameter matrix-level fusion vector. This selection is based on the elements exhibiting the highest variances across multiple target LLMs, as these variances are indicative of the most significant differences and strengths among the models.
> 2. We then compute a matrix-level merging coefficient for each target LLM based on the sum of squares of elements in their respective selected fusion vectors.
> 3. To mitigate knowledge interference across different target LLMs, we implement a conflict resolution mechanism. This entails eliminating elements with minority 	directions when the signs of weight updates are in opposition.
>
> We acknowledge that the SCE merging method involves technical complexity. Due to space constraints in the initial submission, we were unable to elaborate further on its specifics. In the revised version, we will provide a more detailed and comprehensive explanation to ensure clarity and address this complexity effectively.

---

> ### Author Response · Authors · 2024-11-20
> **Official Comment by Authors: Part 2**
>
> > **Q4: Regarding the imbalanced performance achieved by FuseChat in different domains.**
>
> **A4:** We appreciate the reviewer’s concern regarding the imbalanced performance in different domains. The performance of FuseChat across different domains is largely determined by two key factors: **the availability of domain-specific training data and the competency of the source LLMs in those domains**. (The domain distribution of our training data is summarized in the table below.) As shown in Figure 3 of the paper, the pairwise fusion process significantly enhances performance in domains like Math and Coding, where the source LLMs are particularly strong and ample domain-relevant training data is available. In contrast, performance is comparatively lower in domains where the source LLMs are less proficient (e.g., Extraction) or where domain-specific training data is sparse (e.g., STEM). This detailed analysis will be included in the revised version.
>
> | Statistics |  Math | Extraction | Roleplay | Writing | STEM | Reasoning | Humanities | Coding | Total |
> |:--------------:|:-----:|:----------:|:--------:|:-------:|:----:|:---------:|:----------:|:------:|:-----:|
> | Num.Sample     | 15079 | 20329      | 8137     | 7627    | 983  | 7948      | 1403       | 27119  | 88625 |
> | Percentage (%) | 17.01 | 22.94      | 9.18     | 8.61    | 1.11 | 8.97      | 1.58       | 30.60  | 100   |

---

> ### Comment · Area_Chair_EmLf · 2024-11-25
>
> Dear Reviewer,
>
> I noticed that you haven't yet responded to the author's rebuttal. As November 26 is the last day for reviewers to ask questions to authors, could you please review their responses and provide your feedback?
>
> Your timely response will ensure a thorough evaluation process and help with making the final recommendation. Thank you for your prompt attention to this matter.
>
> Area Chair

---

> ### Author Response · Authors · 2024-11-26
>
> Dear Reviewer UpRG,
>
> We sincerely appreciate the time and effort you have devoted to providing thoughtful reviews and valuable feedback. We have carefully addressed your concerns in detail and incorporated additional experiments and analyses, as summarized in the discussion:
>
> - Demonstrated the challenges of knowledge fusion and FuseChat.
> - Detailed the token alignment and SCE merging methods.
> - Explained the imbalanced performance achieved by FuseChat in different domains.
>
> We hope these revisions and discussions have adequately addressed your concerns. As the Author-Reviewer discussion phase is ending soon, we would be grateful for any additional comments or questions that could further enhance our work. Thank you again for your time and effort.
>
> Best regards,
>
> Authors

---

> ### Author Response · Authors · 2024-12-02
>
> Dear Reviewer UpRG,
>
> Thank you for your time and detailed feedback on our manuscript. **As the reviewer-author discussion period ends today (December 2nd at 11:59 pm AoE)**, we would like to check if we have adequately addressed all your concerns.
>
> Your insightful comments and questions have been instrumental in improving our work. **We have carefully incorporated your feedback into the revised manuscript and hope that our responses and updates have successfully addressed all the points you raised.**
>
> We understand you have a busy schedule, but if you have any remaining questions or need further clarification, please let us know, and we will address them promptly. **If you feel that we have satisfactorily addressed your concerns, we would greatly appreciate it if you could consider updating your initial score.**
>
> Thank you again for your valuable time and constructive feedback that has helped enhance the quality of our work.
>
> Best regards,
>
> The Authors of Paper 6352

---

### Official Review · Reviewer_CcuV · 2024-10-30

**Soundness:** 2
**Presentation:** 3
**Contribution:** 2
**Rating:** 5
**Confidence:** 4

**Summary:**

The paper introduces FUSECHAT, a framework designed for the knowledge fusion of chat-based large language models (LLMs). The proposed fuse-and-merge framework integrates the strengths of diverse LLMs through lightweight continual training while avoiding the high cost and potential redundancy associated with developing new LLMs from scratch. Experimental results indicate that the proposed model outperforms existing methods across AlpacaEval and MT-Bench.

**Strengths:**

1.	The motivation is practical and significant, offering a cost-effective solution for integrating capabilities of different heterogeneous LLMs without training new models from scratch.

2. The two-stage framework effectively combines heterogeneous model knowledge through distillation into homogeneous models followed by parameter merging, with a well-designed token alignment strategy.

3. Comprehensive experiments validate the framework's effectiveness, showing competitive performance against different methods.

**Weaknesses:**

1. The paper's technical contribution appears somewhat limited. The approach can be viewed as a combination of pairwise FuseLLM and model merging (similar to TIES-Merging), both of which have been previously established as effective methods. The improved performance, while notable, follows logically from the combination of these known techniques, making the technical innovation less impressive than desired.
2. Several claims in the paper require further clarification. For instance, the statement on line 92 of the Introduction that "FUSELLM limits its exploration to source LLMs of the same size as the target LLM" appears inconsistent with FUSELLM's design, which can handle different-sized source models. Furthermore, FUSECHAT doesn't present special designs for distilling from differently-sized source models. Additionally, the choice of MinCE for the Fusion function in Equation 2 reduces to single-model distillation of the model with lower CE score in each pair, raising questions about the necessity of the pairwise approach.
3. There are concerns regarding experimental details. The combination weight is 0.9 in Equation 4, which means only 0.1 weight is assigned to distillation loss. Compared to 0.9 for SFT, this setting potentially undermines the significance of the distillation process. Moreover, the modest performance difference between FUSECHAT and Pairwise Fusion shown in Table 1 warrants statistical significance testing to validate the improvements.

**Questions:**

1.	Have the authors considered individual model distillation instead of pairwise fusion, given the MinCE choice in Equation 2?
2.	What is the rationale behind the 0.9/0.1 weight distribution in Equation 4?
3.	Can the authors provide statistical significance tests for the improvements over Pairwise Fusion?

---

> ### Author Response · Authors · 2024-11-20
> **Official Comment by Authors: Part 1**
>
> Thank you for reviewing our paper and providing insightful feedback. We're glad you find our work well-motivated and practical. We will address your concerns in the following points.
>
> > **Q1: Regarding the technical innovation and contribution of FuseChat.**
>
> **A1:** We thank the reviewer for the feedback regarding the technical contributions of our work. We would like to emphasize the uniqueness of FuseChat and clarify its distinctions from prior works such as FuseLLM and TIES from multiple perspectives:
>
> **1. Distinction from FuseLLM**
>
> **a. Motivation**
> While FuseLLM emphasizes the fusion of multiple base LLMs through continual pre-training, FuseChat focuses on integrating diverse chat-oriented LLMs into a unified chat model via supervised fine-tuning. This difference in both training objectives and data makes FuseChat essential in the context of chat-focused LLMs. Moreover, our work extends beyond FuseLLM’s scope by fusing six distinct chat LLMs (as opposed to FuseLLM’s three base models), thereby demonstrating the scalability and depth of our methodology.
>
> **b. Methodology**
> While FuseLLM directly employs multi-teacher distillation to fuse multiple base LLMs, FuseChat employs a sophisticated fuse-and-merge approach, beginning with pairwise fusion and advancing to our SCE merging strategy. This new method is not only highly scalable and efficient but also better resolves knowledge conflicts in the parameter space. Simultaneously, it integrates the strengths of each source LLM with precision. By adopting this refined approach, FuseChat noticeably enhances the final model’s performance, distinguishing it from the techniques employed by FuseLLM.
>
> **c. Scalability**
> Another key strength of FuseChat lies in its plug-and-play approach for integrating new LLMs, which is more efficient than FuseLLM. Instead of combining distribution matrices from all source LLMs, FuseChat merges the distribution matrices of the new source LLM with a pivot LLM. This streamlined process reduces computational and storage costs, ensuring superior scalability as the number of LLMs increases.
>
> **d. Experimental Validation**
> Our experimental setup demonstrates the distinct focus of FuseChat. By fusing six varied chat LLMs (OpenChat-3.5-7B, Starling-LM-7B-alpha, NH2-SOLAR-10.7B, InternLM2-Chat-20B, Mixtral-8x7B-Instruct, and Qwen-1.5-Chat-72B), we validate FuseChat on AlpacaEval 2.0 and MT-Bench, assessing both instruction-following and conversational capabilities. This is in contrast to the base-model-focused experiments of FuseLLM and underscores the tailored contributions of FuseChat to the domain of chat LLM fusion.
>
> **2. Distinction from the TIES merging method**
>
> Our SCE merging strategy introduces considerable innovations compared to the TIES merging method:
>
> **a. Automation and Precision**
> Unlike TIES, which relies on manually tuned, model-level coefficients, our SCE automates the merging process by leveraging weight updates from a pivot LLM and computing matrix-level coefficients. This enables the fine-grained incorporation of diverse benefits across LLMs, which is difficult to achieve with manual hyperparameter tuning.
>
> **b. Nuanced Parameter Adjustments**
> In our specific context, where target LLMs are trained on identical datasets with relatively subtle parameter variations, SCE excels at capturing and preserving the distinctive advantages of each LLM through nuanced matrix-level parameter updates.
>
> **c. Superior Performance**
> Experimental results (e.g., Table 4) demonstrate that SCE outperforms baseline merging techniques including TIES within our framework, validating its efficacy and impact.
>
> We will incorporate these detailed discussions into the revised manuscript to provide a clearer distinction of our work from previous approaches.

---

> ### Author Response · Authors · 2024-11-20
> **Official Comment by Authors: Part 2**
>
> > **Q2: Regarding the difference between pairwise fusion and single-model distillation.**
>
> **A2:** Thank you for raising this insightful point. The key distinction between pairwise fusion and single-model distillation lies in their respective learning paradigms. **In pairwise fusion, the model selectively acquires knowledge based on the quality of outputs from the source LLM or the pivot LLM**, guided by lower cross-entropy (CE) values. This mechanism ensures that the model learns from the stronger performer in each sample. In contrast, **single-model distillation relies exclusively on the source LLM, implicitly assuming that the source LLM consistently provides the superior results**.
>
> To address your comment more rigorously, we conducted additional experiments comparing the two approaches. Specifically, we replaced the pairwise fusion strategy in FuseChat with direct distillation from a single (source) model, skipping the merging phase for direct comparison. The results summarized in the table below demonstrate that **pairwise fusion consistently outperforms single-model distillation across five source LLMs**. For clarity, D/P represents the results from direct distillation and pairwise fusion, respectively. Metrics reported include the Average Score on MT-Bench and the Length-Controlled Win Rate on AlpacaEval-2.0.
>
> |              Model             |  MT-Bench | AlpacaEval-2.0 |
> |:------------------------------:|:---------:|:--------------:|
> | OpenChat-3.5-7B Qwen (D/P)     | 6.79/**7.23** | 5.98/**14.98**     |
> | OpenChat-3.5-7B Mixtral (D/P)  | 7.03/**7.24** | 16.10/**16.52**    |
> | OpenChat-3.5-7B InternLM (D/P) | 6.88/**7.21** | 6.54/**15.21**     |
> | OpenChat-3.5-7B SOLAR (D/P)    | 7.09/**7.17** | 12.21/**16.51**    |
> | OpenChat-3.5-7B Starling (D/P) | 7.15/**7.22** | 14.89/**16.20**    |
>
> We further applied our proposed SCE method to fuse the models obtained through single-model distillation. The results below reveal that **merging the models derived from pairwise fusion produces a superior fused model compared to merging models from single-model distillation**.
>
> |       Model       |  MT-Bench | AlpacaEval-2.0 |
> |:-----------------:|:---------:|:--------------:|
> | FuseChat-7B (D/P) | 6.91/**7.38** | 14.68/**17.16**    |
>
> These results highlight the effectiveness of the pairwise fusion approach, not only in standalone performance but also in enhancing the quality of the final fused model. We appreciate your attention to this critical aspect and hope these findings provide additional clarity.
>
> > **Q3: Regarding the rationale behind the 0.9/0.1 weight distribution in Equation 4.**
>
> **A3:** We appreciate the reviewer’s thoughtful observation regarding the rationale behind the 0.9/0.1 weight distribution in Equation 4. This choice is informed by the significant difference in magnitude between the SFT loss and the fusion loss. To further clarify, we conducted a new experiment using Qwen-1.5-Chat-72B as the source LLM and 128 instances randomly sampled from the training dataset. The resulting loss values for SFT and fusion are summarized in the table below.
>
> | Loss Type |  Loss Value  |
> |:---------:|:------:|
> | SFT       | 0.5077 |
> | Fusion    | 1.3081 |
>
> The results show that **the fusion loss is approximately three times larger than the SFT loss** in this experiment. This notable disparity underscores the importance of assigning a proportionally smaller weight to the fusion loss in Equation 4. Without this adjustment, an excessively high weight for the fusion loss could amplify the imbalance, potentially skewing the training process. Thus, the 0.9/0.1 distribution reflects a principled approach to mitigating this effect and achieving a balanced optimization.

---

> ### Author Response · Authors · 2024-11-20
> **Official Comment by Authors: Part 3**
>
> > **Q4: Regarding the statistical significance of performance improvements.**
>
> **A4:** We appreciate the reviewer’s concern regarding the statistical significance of the performance improvements. To address this, we conducted a detailed statistical analysis using t-tests to evaluate the performance of the final FuseChat model against pairwise fusion on MT-Bench. Moreover, we performed a similar analysis comparing the final FuseChat model with OpenChat-3.5-7B Multi, which integrates multiple source LLMs simultaneously, as FuseLLM does. The results summarized in the following table demonstrate **the strong statistical significance of the final FuseChat model's superiority over these baselines**. These results will be included in the revised paper to enhance clarity and provide robust support for our claims.
>
> |                 Model                 | t-statistic | p-value |
> |:-------------------------------------:|:-----------:|:-------:|
> | FuseChat-7B vs. Pairwise Fusion       | 2.95874     | 0.00318 |
> | FuseChat-7B vs. OpenChat-3.5-7B Multi | 3.32756     | 0.00108 |
>
> > **Q5: Regarding claims that require further clarification.**
>
> **A5:** We sincerely thank the reviewer for highlighting this important point. Regarding the claim about the limitations of FuseLLM, we wish to clarify that while FuseLLM's experiments were constrained to three source LLMs of an equivalent 7B scale, our work broadens the scope by incorporating six source LLMs with varying scales, ranging from 7B to 72B. We will ensure that these claims are conveyed more clearly in the revised manuscript.

---

> > ### Comment · Reviewer_CcuV · 2024-11-25
> > **Thank you for responding to my concern.**
> >
> > Thank you for responding to my concern, but I feel that there is still a slight lack of innovation. I plan to keep my score unchanged.

---

> > > ### Author Response · Authors · 2024-11-25
> > >
> > > Thank you for carefully considering our response to your concerns. We appreciate your feedback and are pleased to hear that our revisions addressed many of the points you raised. We fully respect your judgment on the novelty of our work. However, we hope you can understand that, given the generally favorable ratings from the other two reviewers (8 and 6), your score of 5 may significantly influence the likelihood of our paper's acceptance.
> > >
> > > Since we seem to have common ground on the motivation behind our work, the proposed two-stage framework, the token alignment strategy, as well as the experimental results, we would kindly ask if you might reconsider raising your score from 5 to 6. This adjustment would greatly enhance the possibility of our work being presented to a broader audience at ICLR.
> > >
> > > Thank you for your time and thoughtful consideration.

---

### Official Review · Reviewer_qdGR · 2024-11-01

**Soundness:** 3
**Presentation:** 4
**Contribution:** 3
**Rating:** 8
**Confidence:** 4

**Summary:**

This paper proposes a new framework, FuseChat, to fuse diverse LLMs into a single LLM capable of performing various tasks.  They first apply pairwise knowledge fusion on source chat LLMs to create multiple target LLMs with identical structures. To fuse models with different vocabulary, they introduce a statistics-based token alignment approach to obtain probabilistic distribution matrices. Then, they fuse all the target LLMs within the parameter space by utilizing the proposed new merging method, SCE. In their experiments, they conducted extensive experiments to investigate their framework with diverse source LLMs and evaluation datasets. They also offered a number of model analyses and ablation studies.

**Strengths:**

1. The paper studies an interesting question of how to fuse multiple chat LLMs into a potent chat LLM. The paper is well-written and well-organized.
2. The paper has extensive experiments to investigate the effectiveness of their proposed framework and each component in their framework.
3. Their fusion method is also computation-friendly, which doesn't require additional training or dataset.

**Weaknesses:**

1. They didn't provide a significance test to show if their proposed method significantly outperforms their baselines (e.g. FuseLLM/OpenChat-3.5-7B Multi) or not. Because the improvement in some tasks is small, it would be better to show whether the improvement is significant.
2. Table 1's caption needs to be improved. It would be helpful if they clarified what bold font and underscore mean in their table and what the percentage means.

**Questions:**

1. For Figure 3, I wonder if pivot LLM is the original OpenChat-3.5-7B or OpenChat-3.5-7B after fusing training. I also wonder if Target LLM is the OpenChat-3.5-7B after fusing training or the final FuseChat model. Please clarify these.
2. I wonder if you could categorize the samples in your training data into domains by MT-Bench and see how the distribution is in your training set.

---

> ### Author Response · Authors · 2024-11-20
> **Official Comment by Authors**
>
> Thank you for reviewing our paper. We greatly value your insightful feedback and appreciation of our work's significance. Below, we address your concerns in detail.
>
> > **Q1: Regarding the significance test in our experiments.**
>
> **A1:** Thank you for raising this important point. To show the statistical significance of our results, we conducted a t-test on MT-Bench to compare the performance of our proposed FuseChat-7B with OpenChat-3.5-7B Multi, which fuses multiple source LLMs simultaneously. The results shown in the table below reveal a p-value well lower than 0.05. This confirms that **FuseChat-7B achieves statistically significant improvements over OpenChat-3.5-7B Multi**. These statistical results will be incorporated into the revised manuscript.
>
> | Model | t-statistic | p-value |
> |:-----------:|:-----------:|:-------:|
> | FuseChat-7B vs. OpenChat-3.5-7B Multi | 3.32756     | 0.00108 |
>
> > **Q2: Regarding the caption of Table 1.**
>
> **A2:** Thank you for your thoughtful suggestion. We will address this in the revised version by improving the caption for Table 1. Specifically, we  will clarify that the bold font denotes the best performance among all the fused LLMs, while the underscore indicates the second-best performance. Moreover, the percentages represent the relative performance improvement compared to the OpenChat-3.5-7B SFT baseline model. We believe these clarifications enhance the table’s interpretability and precision.
>
> > **Q3: Regarding the details of pivot LLM in Figure 3.**
>
> **A3:** Thank you for your observation. As clarified in Section 3.1 (lines 179–190), the term pivot LLM refers to the original OpenChat-3.5-7B model prior to the application of pairwise fusion, and the term target LLM describes an intermediate model generated through pairwise fusion between the pivot LLM and an individual source LLM. Our approach first performs pairwise fusion between the pivot LLM and each source LLM independently, resulting in a series of corresponding target LLMs. These intermediate models are then combined using our SCE merging technique to create the final FuseChat model. We hope this explanation resolves any potential ambiguity.
>
> > **Q4: Regarding the domain distribution of samples in training data.**
>
> **A4:** We appreciate the reviewer’s concern regarding the domain distribution of samples in the training data. We followed the approach described in Magpie [1] and employed the Llama-3-8B-Instruct model to classify our 95,000 training examples into eight distinct domains as defined by MT-Bench. After excluding approximately 7,000 samples due to anomalous classification errors, the final domain distribution is summarized in the following table:
>
> | Statistics |  Math | Extraction | Roleplay | Writing | STEM | Reasoning | Humanities | Coding | Total |
> |:--------------:|:-----:|:----------:|:--------:|:-------:|:----:|:---------:|:----------:|:------:|:-----:|
> | Num. Sample    | 15079 | 20329      | 8137     | 7627    | 983  | 7948      | 1403       | 27119  | 88625 |
> | Percentage (%) | 17.01 | 22.94      | 9.18     | 8.61    | 1.11 | 8.97      | 1.58       | 30.60  | 100   |
>
> The resulting data distribution demonstrates substantial diversity, which aligns with our primary objective to assess the model's general capabilities rather than domain-specific performance. As shown in Figure 3 of the paper, pairwise fusion leads to a marked improvement in the target LLMs' math and coding abilities. This enhancement is primarily due to **the strong performance of the source LLMs in these domains, coupled with the relatively high proportion of Math and Coding samples in our dataset.** Interestingly, despite a notable representation of the Extraction domain in the dataset, the target LLMs show limited improvement in this area. This can be attributed to **the relatively weaker performance of the source LLMs in extraction tasks, highlighting the critical role of selecting appropriate source LLMs for domain-specific objectives.** We will integrate this detailed analysis in the revised manuscript to provide further clarity.
>
> [1] Xu et al. Magpie: Alignment Data Synthesis from Scratch by Prompting Aligned LLMs with Nothing, 2024.

---

> > ### Comment · Reviewer_qdGR · 2024-11-23
> > **Thanks for your response.**
> >
> > I found that my concerns have been addressed. I have adjusted my rate accordingly.

---

> > > ### Author Response · Authors · 2024-11-28
> > > **Thanks**
> > >
> > > Thanks for your positive feedback and for the constructive comments that are pivotal to improve our work.

---

### Author Response · Authors · 2024-11-28
**Major Revisions of the Manuscript**

Dear Reviewers,

We sincerely appreciate your thorough review and valuable feedback, which have greatly contributed to improving the quality of our manuscript. We have carefully addressed all comments and suggestions through comprehensive revisions. Below, we summarize the major changes made to the manuscript, with key updates highlighted in ${\color{blue} blue}$ text in the PDF, along with other refinements to meet the page limit.

> **Key Revisions**

1. We have refined the discussion regarding FuseLLM's limitations in **Section 1 (Lines 92–93)** for enhanced precision. (Reviewer **CcuV**)

2. We have updated the caption for **Table 1** for improved clarity. (Reviewer **qdGR**)

3. We have added an explanation regarding the model's performance differences across various domains in **Section 4.2 (Lines 432–434)**, indicating the performance is largely determined by the availability of domain-specific training data and the competency of the source LLMs in those domains. (Reviewers **qdGR, UpRG**)

4. We have included the domain distribution of training data across different domains in **Appendix C (line 852-866), Table 6**. (Reviewer **qdGR**)

5. We have added significance tests for the performance improvements of FuseChat over Pairwise Fusion and OpenChat-3.5-7B Multi in **Appendix F (line 929-942), Table 8**, demonstrating the strong statistical significance of the final FuseChat model's superiority over these baselines. (Reviewers **qdGR, CcuV**)

6. We have added an analysis comparing Pairwise Fusion and Single-Model Distillation in **Appendix G (line 945-971), Table 9, Table 10**, , establishing the consistent superiority of pairwise fusion across five source LLMs and the enhanced effectiveness of merging pairwise fusion models versus single-model distillation models. (Reviewer **CcuV**)

7. We have elaborated on the rationale behind the large weight of fusion loss in **Appendix H (line 972-986), Table 11**, explaining its relationship to the approximately threefold magnitude compared to the SFT loss. (Reviewer **CcuV**)

We are deeply grateful for your insightful feedback, which has been instrumental in strengthening our work. We hope these revisions and additional analyses thoroughly address all concerns. As the Author-Reviewer discussion phase nears its conclusion, we welcome any further suggestions that could help us improve the manuscript.

Best regards,

Authors

---

### Note · Authors · 2025-01-23

I have read and agree with the venue's withdrawal policy on behalf of myself and my co-authors.